# Assessing the Uniformity in Australian Animal Protection Law: A Statutory Comparison

**DOI:** 10.3390/ani11010035

**Published:** 2020-12-26

**Authors:** Rochelle Morton, Michelle L. Hebart, Rachel A. Ankeny, Alexandra L. Whittaker

**Affiliations:** 1School of Animal and Veterinary Sciences, The University of Adelaide, Roseworthy, SA 5371, Australia; michelle.hebart@adelaide.edu.au (M.L.H.); alexandra.whittaker@adelaide.edu.au (A.L.W.); 2School of Humanities, The University of Adelaide, Adelaide, SA 5005, Australia; rachel.ankeny@adelaide.edu.au

**Keywords:** animal welfare legislation, animal cruelty, law enforcement, Australia, enforcement gap

## Abstract

**Simple Summary:**

Australia does not have any federal legislation pertaining to animal welfare; thus, the responsibilities lie with each state and territory. This situation has led to eight different pieces of animal welfare legislation across the country, with potentially distinct content and avenues for interpretation. These differences may create problems for the enforcement of animal welfare law, and hence it has been suggested that a uniform approach is required. However, before such an approach can be considered, the extent of the inconsistencies between the states and territories needs to be assessed. This review compares the differences between state and territory animal welfare laws to determine the presence and nature of any major inconsistencies. A total of 436 primary pieces of legislation were reviewed, with 42 included in the detailed analysis. Animal welfare laws were found to be generally consistent across the states and territories of Australia, but with some important shortcomings that are discussed.

**Abstract:**

Animal welfare is not included in the Australian Constitution, rendering it a residual power of the states and territories. Commentators have suggested that inconsistencies exist between the state and territory statutes, and that a uniform approach would be beneficial. However, there has been no comprehensive assessment of the nature or extent of these purported inconsistencies. This review addresses this gap by providing a state-by-state comparison of animal protection statutes based on key provisions. Utilizing systematic review methodology, every current Australian statute with an enforceable protection provision relating to animal welfare was identified. A total of 436 statutes were examined, with 42 statutes being included in the detailed analysis. The comparison showed that animal protection laws are generally consistent between each Australian jurisdiction and were found to have similar shortcomings, notably including lack of a consistent definition of ‘animal’ and reliance on forms of legal punishment to promote animal welfare which have questionable effectiveness. It is argued that there is a need for attention to definitions of key terms and future consideration of alternative forms of penalties, but that a uniform federal approach may not be necessary to address these shortcomings.

## 1. Introduction

In Australia, there is no overarching federal animal protection legislation, as there is no ‘head of power’ for animal welfare in the *Commonwealth of Australia Constitution Act 1901* (‘Constitution’) [1]. Due to this lack of Constitutional recognition, animal welfare is a residual power within the domains of the Australian states and territories. Consequently, animal protection laws have been individualized for each state and territory and, as a result, it has been argued that this causes cross-jurisdictional inconsistencies [2,3,4,5]. These inconsistencies are thought to create a “fragmented, complex, contradictory, inconsistent system of regulatory management” [3]. Some of the criticisms of a state-based approach in Australia and elsewhere include that it makes national data collection almost impossible [3], causes public confusion [4], does not allow for cross-jurisdictional recognition of animal prohibition orders [6] and does not present a united front toward animal protection [2].

The shortcomings around lack of harmonization of animal welfare laws have previously been discussed in the Australian context [2,3,5,6] and in Canada [4]. It has been argued that a more uniform approach to animal welfare laws would be beneficial [2,3,4,6], and that these types of inconsistencies can contribute to an ‘enforcement gap’ in animal law [6], previously defined as a discrepancy between the intentions of the written law and the outcomes of the enforcement process [6,7], where intentions do not align with outcomes. It is important to identify shortcomings in animal welfare legislation, isolate key issues, and propose targeted reform. However, in spite of the ongoing discourse on this topic, there has not been an extensive assessment of the existence and extent of inconsistencies between current Australian state and territory legislation pertaining to animal welfare.

This paper hence seeks to provide a legislative review relating to governance of animal welfare that can serve as a basis for guiding future discussions around need for a uniform approach to animal welfare law. Utilizing systematic review tools, we examine every current Australian statute which has an enforceable protection provision relating to animal welfare, where ‘animal welfare’ is defined as promoting duty of care responsibilities towards animals, as well as preventing cruel acts towards them that result in harm or suffering. It is important to note that this study only focuses on statutes. Additionally, delegated legislation such as regulations and codes of practice are not included and should be considered for future review. The resulting analysis is designed to provide easy access to state-by-state comparisons and is not intended to provide full discussion on the pros and cons of all statutory inclusions. Consequently, this review will discuss the legal implications of common definitions, such as ‘animal’, ‘welfare’ and ‘cruelty’, compare the differences between the animal welfare provisions cross-jurisdictionally and conclude with potential avenues for reform.

## 2. Materials and Methods

### 2.1. Data Sources and Search Strategy

The legislation search was conducted in the LawOne by TimeBase electronic database, in February 2020. The search was limited to current Australian statutes, including both national (Commonwealth) statutes, and state and territory-based statutes. Thus, any repealed acts, bills, regulations or codes of practices were not included in the search. The search strategy was developed in consultation with a legal information specialist. Common terms used in animal protection statutes were identified and used to create the search terms. The search criteria were as follows: animal OR livestock OR wildlife OR fish AND welfare OR protection OR cruelty OR bestiality OR harm OR injury.

### 2.2. Eligibility Criteria

Statutes accepted for analysis included any with provisions for the protection of animal welfare or prevention of animal cruelty as defined by promoting duty of care responsibilities towards animals or preventing cruel acts towards them that result in harm or suffering, and that were enforceable by penalty. Penalty was defined broadly to include monetary fine, custodial sentence, animal welfare directions or notices, court-mandated prohibitions or animal seizures. Each statute was reviewed manually using key search terms of the inclusion criteria. For inclusion, provisions had to have the protection of animal welfare as their primary object, rather than animal welfare protection that arises as a result of achieving the primary purpose (see below).

Statutes were excluded from the analysis if they made reference to animal welfare in the absence of an associated penalty. Examples include referral in object clauses of statutes, reference to relevant delegated legislation, and references to other statutes (most commonly state and territory-based animal welfare statutes). Statements that animal welfare experts should be included on committees were also excluded, as were any statutes put in place for the management of animals largely for public health purposes (e.g., dog and cat management acts), and any statute controlling humane killing methods (e.g., biosecurity acts). Provisions that included stealing or killing animals with the intention to steal were excluded (sections of crimes acts), because they relate to damaging personal property and are not focused on animal welfare. Statutes in force to regulate professions, such as the veterinary industry or research involving animal use, were excluded as they are largely administrative in nature (e.g., controlling licensing) and lack any enforceable animal welfare provisions. Similarly, emergency management statutes that could make reference to managing animals in natural disasters (see [8] for further commentary) were excluded for the same reasons. Finally, any statutes that discussed the human effects of animal abuse were excluded (e.g., domestic violence acts), as these acts are in place to protect human suffering resulting from emotional abuse caused by animal cruelty.

### 2.3. Data Extraction

This paper does not review the statutes in their entirety but focuses on the provisions in those with an animal welfare component. Variables of interest included definitions of common terms, such as ‘animal’, and welfare provisions, such as offences and their corresponding penalties, as well as court orders and animal welfare directions.

## 3. Results

A broader search using the term “animal” identified 3529 potentially relevant statutes. The search was then limited to current acts, resulting in 3093 regulations, bills, assented, amending and repealed acts being excluded. A manual full-text review of the 436 remaining statutes was conducted and a further 394 acts were excluded as they did not include enforceable provisions for animal welfare. Thus, a total of 42 statutes (refer to Appendix A for list) were included in the statutory analysis (Figure 1).

### 3.1. Type of Statutes

The 42 statutes from nine Australian jurisdictions included in the analysis can be broken down into eight different types of animal protection categories (Table 1). Each act has specific categories of animal use that it protects. Each state and territory has an ‘animal welfare act’, where the focus is to protect all categories of animals as per the specific definition outlined in the act. Each state and territory also has ‘crimes acts’ which include specific offences relating to animals. The only Commonwealth acts included in the analysis are wildlife acts, which include protection provisions for specific types of Australian wildlife. The majority of states and territories have fisheries acts that are in place for fishery management. The remaining statutes occur non-uniformly across the states and territories and include two livestock acts with protection provisions, two animal sporting acts, one exhibited animals act and a variety of miscellaneous acts. The miscellaneous acts include those for chemical use, summary offences and police powers, which all have at least one offence relating to protection of animal welfare (refer to Appendix A).

### 3.2. Animal Welfare Statutes

There are eight separate animal welfare statutes in place at the state and territory level in Australia (Table 2). The overarching objective of these statutes is to prevent animal cruelty by promoting animal welfare [10,11,12,13,14,15,16,17]. They are the primary pieces of legislation in place to define, penalize and deter acts of animal cruelty [6]. The following discussion concentrates on provisions deemed by the authors to be of central importance for discussions about uniformity and harmonization.

#### 3.2.1. Definition of ‘Animal’

Definition of the term ‘animal’ in animal welfare statutes is generally consistent; however, there are some differences between each state and territory statute based on animal subgroups (orders and classes, see Table 3). All states and territories include mammals, reptiles, amphibians and birds in their definitions of an ‘animal’. Inconsistencies arise for aquatic species, such as fish, crustaceans and cephalopods, as many states and territories do not include them under the definition of an ‘animal’. In some states and territories (ACT, NSW and NT), there are specific provisions in place to include certain fish or crustaceans based on their use by humans, namely as captive fish (NT) or crustacea used for human consumption (ACT and NSW). All states and territories, aside from VIC [26] and QLD [27], make no mention of animals in prenatal forms in their definitions.

#### 3.2.2. Definition of ‘Owner/Person in Charge’

The definition of who is deemed to be in charge of an animal is important for animal law enforcement. The terms ‘owner’ or ‘person in charge’ are used throughout state and territory animal welfare acts. The terms ‘custody’ and ‘control’ are ‘used consistently across jurisdictions (Table 4), whereas other less common terms include ‘care’ and ‘proprietary interests’ (the latter only included in the QLD definition).

#### 3.2.3. Animal Welfare Offences

Each animal welfare act has varying offences relating to protecting animal welfare and preventing animal cruelty (Table 5). There are three main offence types, being ‘duty of care breaches’, ‘basic cruelty’ and ‘aggravated cruelty’. The aggravated offence is more serious since it results in serious harm to, or death of the animal. Some statutes also include the need to prove *mens rea* (the mental element) in the aggravated offence [37], whereas others have no inclusion of *mens rea*, such in the case of Victoria [38] and NSW [39], making them strict liability offences [40]. In the former there is a need to prove that the defendant intended to, or was reckless about, causing a high degree of suffering or death of an animal, where all elements of the offence can be proven beyond reasonable doubt. In contrast, strict liability offences, such as basic cruelty offences, do not require proof of intent or recklessness. They are determined by the court based on an objective standard, where the court will consider whether the ‘ordinary, reasonable person in the defendants circumstances’ would have acted similarly [41]. It is recognized that animal welfare offences can occur from either an omission to act (e.g., animal neglect such as failure to provide veterinary care) or commission of an act (e.g., animal abandonment).

Each of the offences are described in terms of culpability and severity of harm caused to the animal, and consequently have different maximum penalties. Penalties are generally listed as custodial terms or monetary fines. Duty of care breaches generally attract lower maximum penalties compared to aggravated offences. Not all states and territories distinguish all three types of offences in their legislation. For example, QLD does not outline an aggravated offence in their *Animal Care and Protection Act 2001*, and SA includes both duty of care breaches and basic cruelty offences under the singular Section 13(2) of their *Animal Welfare Act 1985.* Some states and territories also have sections in their acts that outline specific offences, such as ACT where ss6B-G of the *Animal Welfare Act 1992* outlines specific duty of care breaches, and NSW in ss8-11 of the *Prevention of Cruelty to Animals Act 1979.* Every state and territory aside from SA use penalty units to describe fines, whereas SA uses dollar amounts. Penalty units have been implemented as a simple way to increase the dollar value of a fine in line with public policy or inflation. A separate sentencing act defines the monetary value of a penalty unit, hence only a single amendment to the sentencing act is required to increase dollar amounts across a breadth of legislation [42]. Only WA includes minimum monetary penalties totaling $2000 for the basic animal welfare offence under Section 19(1) of the *Animal Welfare Act 2002.* All states include maximum custodial sentences in offence provisions.

Each animal welfare statute prescribes a range of prohibited activities, such as animal fighting, as well as use of prohibited items, such as traps and electrical devices (Table 6). These offences are presented in a generally consistent manner. However, there are inconsistencies apparent within the types of activities and items prohibited in each jurisdiction. This is likely due to the relationship between the statutes relevant delegated legislation, in the forms of regulations and codes of practices.

#### 3.2.4. What Constitutes Cruelty?

The definition of ‘cruelty’ is relatively consistent between jurisdictions (Table 7). Most states and territories define cruelty variously as causing an animal unnecessary, unreasonable or unjustifiable pain, harm or suffering, where pain, harm or suffering is defined as forms of distress and injury to animals. However, all statutes have subtle differences in presenting the definition. To further explain, states such as ACT define cruelty specifically under a definitions clause [76], whereas QLD includes examples of behaviors which constitute cruelty under the cruelty offence [52]. All jurisdictions, aside from Victoria, follow this same mechanism of either defining cruelty specifically or including examples that would meet the definition of cruelty. Conversely, Victoria only includes examples of what constitutes an offence and lacks any cruelty definition. However, in a 2019 report on animal cruelty offences, the Victorian Sentencing Advisory Council interpreted the definition of cruelty under the Victorian Act as “any act or omission that contributes to an animal experiencing, or being likely to experience, unreasonable or unnecessary pain or suffering” [77], which is consistent with the other jurisdictions. Recently the Victorian Government have proposed a series of amendments to their current *Prevention of Cruelty to Animals Act 1986*, in which they have stated “listing specific actions or behaviors can be limiting, as not every specific example is clearly covered” [78] in regard to their current approach of outlining ‘cruelty’. Thus, it is probable that Victoria may take a similar approach to defining cruelty as the other jurisdictions. Based on these definitions, it is likely that there is a significant role for the courts in applying statutory interpretation principles, relying on precedent, and interpreting expert evidence in these matters.

#### 3.2.5. Court Orders

Animal welfare acts outline the types of court orders specific to animals that can be issued for an animal welfare offence (Table 8). Courts additionally have broad powers to issue other relevant orders by virtue of the relevant legislation. There are four types of orders: prohibition, supervision, interim and recognition of interstate orders. Prohibition orders prohibit offenders from owning animals for a set period or until a further order is approved by the courts, or limit the number of animals to be owned [80,81,82,83,84,85,86,87]. Each state and territory’s act grants the power to issue prohibition orders, but the remaining three orders are only provided for sporadically in the acts between the states and territories. Supervision orders, which are enforceable in SA [88] and VIC [89], work in a similar way to prohibition orders. However, instead of offenders being prohibited from owning animals, they are placed under the supervision of an enforcement authority who is empowered to regularly check on the welfare of offenders’ animals. Interim orders are in place in ACT [90] and QLD [91] and can be issued during case investigation to restrict animal ownership. The jurisdictions of NT [92], TAS [93] and VIC [94] also recognize orders made by interstate courts.

### 3.3. Crime Statutes

Unlike animal welfare acts where the focus of the statute is entirely on promoting animal welfare and preventing animal cruelty, crimes acts focus on a wide array of criminal activities and some include serious offences against animals. These acts were enacted to consolidate the common law relating to crimes and criminal offenders [95,96,97,98,99,100,101,102]. There are eight separate crimes acts in place at the state and territory levels (Table 9).

Crimes acts often include the most serious animal welfare offences and attract higher maximum penalties as they are indictable offences, in comparison to animal welfare acts, where commonly (although not exclusively) offences are described as summary (Table 10). As they are more serious in nature, indictable offences may be tried before a jury and may be heard in a higher court. Summary offences are minor offences and are heard in the Magistrates Court [111]. These indictable offences can be categorized as ‘aggravated cruelty’, acts of ‘bestiality’ or offences specific to ‘working animals’. The nature of aggravated offences in the aforementioned animal welfare acts are similar to those in the crime acts. Further, if an animal welfare act does not specifically outline an aggravated offence (e.g., in the case of QLD, see Table 5), an aggravated offence is included in the crimes act (Table 10). However, in WA, there is no specific inclusion of an aggravated offence in either animal welfare or crimes acts. Factors of aggravation which increase culpability, such as committing the offence in company, may however be taken into account when sentencing via provisions of the *Sentencing Act 1995 (WA)* [112]. Maximum penalties for aggravated offences in some jurisdictions can be dependent on specific elements, such as the species or utility of the animals affected in the case of QLD [113], or the mental element (such as intent) of the defendant in NSW [114].

Bestiality is an offence included in all state and territories crimes acts and has the highest maximum penalty of all the offences examined, ranging from three years imprisonment (NT) to 14 years imprisonment (NSW), with no potential for monetary penalties in lieu of custodial sentences. Protection for working animals is included in NSW and SA crimes acts, where working animals are defined as animals used for law enforcement (NSW) [115], and police dogs, police horses, correctional services dogs or accredited assistance dogs (SA) [116]. These offences attract similar maximum penalties (5 years imprisonment) to the closely related aggravated offences in both the crimes acts and animal welfare acts.

### 3.4. Wildlife Statutes

Wildlife statutes have conserving and protecting environments as their primary objective [127,128,129,130,131,132,133,134,135,136,137], and include native Australian animals. Eleven statutes with animal protection provisions for native Australian wildlife were sourced for this analysis (Table 11), two of which are Commonwealth acts (national laws governing the entirety of Australia) and the remaining nine are from the eight state and territory governments, with two statutes included in NSW. Some wildlife protection provisions are included in the Commonwealth *Environment Protection and Biodiversity Conservation Act 1999 (CTH)*, which ratifies the CITES convention domestically [138], and prevails over the state and territorial statutes where there is inconsistency [139].

In total, there are 29 offences described in the 11 statutes relating to animal protection, with a majority of the offences including injuring or harming protected species as defined by the statutes. There are separate offences and maximum penalties for different categories of wildlife: offences against endangered species attract higher penalties than those against vulnerable or unclassified species [140,141,142,143,144,145,146,147,148,149]. As wildlife acts primarily focus on native Australian wildlife, pest species are generally not protected.

### 3.5. Fisheries Statutes

Fishery acts are in force for the conservation of native fish species and habitats, as well as management of commercial fisheries and recreational fishing [180,181,182,183,184,185]. There are only six fishery acts that include protection provisions across six of the states and territories (Table 12). QLD and WA do not provide any protection provisions based on the definition in their relevant fisheries acts. Offences mostly relate to preventing the disturbance of fish and regulating the use of fishing systems that may be detrimental to welfare.

### 3.6. Miscellaneous Statutes

Nine statutes from six states and territories have animal protection provisions that do not fit in any of the aforementioned categories. These statutes range from livestock acts, racing acts, chemical use acts, exhibited animal acts and police powers and summary offences acts (Table 13). Each of these acts have a specific section relating to protecting animals, despite this not being the main objective of the act.

The Tasmanian *Animal Farming (Registration) Act 1994* and Victorian *Livestock Management Act 2010* both have a single animal protection offence, relating to injuring of prescribed species in the former and acting in a way that causes serious risks to animal welfare in the latter. The two animal racing acts are the *Racing Integrity Act 2016* (QLD) and *Racing Act 1958* (VIC). These Acts provide protection to racing animals as well as deterring the use of live animals as lures. Other acts identified as having animal protection provisions were the *Radiation Control Act 1990* (NSW) and the *Agricultural and Veterinary Chemicals (Control of Use) Act 2004* (NT) (see details in Table 13). The QLD *Exhibited Animals Act 2015* allows inspectors to seize any exhibited animal (such as zoo animals) if it is in the best interest of the animal’s welfare. Finally, the *Police Powers and Responsibilities Act 2000* (QLD) provides police officers with greater power to protect animal welfare, and the *Summary Offences Act 1953* (SA), outlines the offence of trespassing and disturbing animals used for primary production.

## 4. Discussion

This review outlines key provisions in Australian law relating to animal protection. This analysis is intended to provide the basis for future discussion on the potential rationale for and desirability of reforms to increase uniformity of animal protection legislation, as is frequently proposed by commentators [2,3,4,5]. We used a restricted definition of animal welfare/protection, namely legislation that seeks to promote duty of care responsibilities towards animals or prevent cruel acts towards them that result in harm or suffering. It should be noted that this review singularly focused on statutes and excluded delegated legislation. Statutes do not work independently [220], instead they rely on a symbiotic relationship between delegated legislation in the forms of regulations and codes of practices, as well as common law principles such as the doctrine of precedent. Without the inclusions of such principles, this review only provides a limited viewpoint and this should be acknowledged as a limitation. Further analysis is required into each component of the legal framework in order to achieve a comprehensive understanding of these shortcomings, this review is the beginning of this process. The remainder of this paper will discuss selected aspects of our analysis with a focus on the implications of the legal framework and potential avenues for reform.

### 4.1. What Is an ‘Animal’?

#### 4.1.1. Defining the Term

The definition of ‘animal’ is a key element in animal protection legislation, as it determines the applicability of the legislation [37]. It could be expected that this definition is underpinned by available scientific evidence, particularly about various animal species’ sentient abilities [221,222,223,224]. It is therefore surprising that there are notable differences between the states and territories (Table 3). The term ‘animal’ is often defined in animal welfare statutes as any member of the vertebrate species, excluding human beings [37]. However, inconsistencies between jurisdictions arise for fish, crustaceans, cephalopods and unborn animals. These inconsistencies suggest that established scientific principles and evidence are not guiding black letter law.

Science can provide information on animals’ sentient abilities [222,223,224,225,226], where ‘sentient’ refers to experiencing an array of feelings from both negative states (pain and suffering) to positive states (pleasure and joy) [227]. It has been claimed by some scholars that an animal’s capacity to experience both negative and positive emotions has been the driving factor for the animal welfare movement and consequential animal protection legislation [222,223]. Thus, it is notable that many laws neglect to acknowledge or discuss animal sentience when defining ‘animals’. The relationship between science and law is important, as science can influence the formation of legislative objectives [228] and provide models for delegated legislation [229,230], and scientific evidence can be critical in court proceedings [231]. As stated by Proctor, Carder and Cornish [224] there is “overwhelming evidence of animal sentience”, with confidence that most vertebrates have sentient abilities [222,223,224], but there is lack of consensus associated with fish [232,233], invertebrates [234,235,236] and fetal or immature forms of various animals [237,238,239,240]. This statement roughly aligns with the position taken at law, as only some jurisdictions recognize fish, crustacea, cephalopods and fetuses as animals (Table 3). However, there is growing evidence in support of fish sentience [241,242,243], and consequently increasing international support for their recognition and protection under animal welfare statutes [244]. Furthermore, fetuses whose age is beyond half the gestation period have also been acknowledged as having sentient abilities and awarded protection under the delegated legislation for animal research, the Australian Code for the Care and Use of Animals for Scientific Purposes [245]. It is postulated that enforcement of animal welfare offences for certain species may be affected by a pragmatic approach related to the ability to enforce provisions, rather than a strict grounding in scientific evidence alone, which creates inconsistencies. For example, enforcing welfare provisions for aquatic species outside of captivity is difficult to achieve due to the vastness of their natural habitats, and the pervasiveness of certain activities such as recreational fishing [246,247]. Thus, this causes the states and territories, such as ACT, NSW and NT, to set parameters in place where fish or crustacea are only included as ‘animals’ if they are captive (NT) [30] or used for human consumption (ACT and NSW) [28,29].

It should be noted that legal definitions are not, and should not, be entirely dictated by scientific evidence, but there will also be a dependence on the corresponding legislature’s objective and the broader social and cultural context in which they occur. For this reason, differences between definitions will be observed between legislation in different subject domains, such as animal welfare and wildlife statutes, as these statutes have been drafted for different purposes. Whilst animal welfare statutes are enacted to promote welfare and prevent cruelty [10,11,12,13,14,15,16,17], wildlife statutes are in force to conserve and protect Australian fauna and flora [127,128,129,130,131,132,133,134,135,136,137]. Thus, although animal welfare statutes define ‘animals’ as any member of the vertebrate species excluding human beings [37], wildlife statutes have a different definition, namely, ‘any member, alive or dead, of the animal kingdom (other than a human being)’ [248]. Deceased animals are included in this definition as their genetics can still be conserved. Based on the wildlife acts, there are some inconsistencies between this definition in jurisdictions, such as the exclusion of invertebrates in ACT [249] despite being a part of the Kingdom Animalia [250], whilst NSW includes invertebrates but excludes fish [251], despite including fish in their definition under the animal welfare statute [29]. Finally, there is a single inclusion of protistans in NT [252], which is a group of organisms not included in the animal kingdom, and thus strictly speaking are not ‘animals’ [250]. Some definitions also include deceased animals as well as unborn animals and reproductive material (eggs) in addition to intact organisms [248,249,251,252,253,254,255,256,257].

The definition of ‘animal’, particularly in animal welfare statutes, requires some revision given the inconsistencies with regard to evidence about animal sentience. Arguably if an animal has sentient abilities it should be awarded protection by law as it can experience suffering. However, it is noted that legal definitions are multifaceted and simply amending the definition to reflect the available scientific evidence is not possible in the sense that it must coincide with the legislative objective and take into account the consequential enforcement.

#### 4.1.2. Legally Recognizing Animal Sentience

Animal sentience can be formally recognized in law. The Australian Capital Territory (ACT) has the most inclusive definition of an ‘animal’ and was the first Australian territory (or state) to reform legislation to formally acknowledge animal sentience through its amendment in 2019 [258] of Section 4A the Animal Welfare Act 1992 to include the following statement:(1)The main objects of this Act are to recognize that—
(a)animals are sentient beings that are able to subjectively feel and perceive the world around them; and(b)animals have intrinsic value and deserve to be treated with compassion and have a quality of life that reflects their intrinsic value; and(c)people have a duty to care for the physical and mental welfare of animals [10].


However, in this context, legally recognizing animal sentience appears somewhat of a symbolic gesture [259], given that the nature of animal welfare offences and accordance with Australian codes of practices does not significantly change as a result. Codes of practices take the form of detailed guides made, or adopted under, animal protection statutes [260]. Codes are controversial in animal law jurisprudence with commentators asserting that they allow the exploitation of animals by providing exemptions to animal welfare offences for industries such as animal farming and research [260,261,262]. However, recognition of sentience in law may still have some positive impacts. The European Union has acknowledged animals as ‘sentient beings’ since the enactment of the Treaty of Lisbon in 2007 [263], and it has been reported that this formal status has assisted in legislative interpretation [264] as well as motivating further legislative protections for animals [265]. Such impacts may occur in ACT, guided by Australian common law, and statutory interpretation principles [220,266], but outcomes of this change are likely to take years to become apparent. Recently, legally recognizing animal sentience was included in the proposal released by the Victorian Government for the amendments to the *Prevention of Cruelty to Animals Act 1986*, where they stated “science tells us that animals are sentient” and “many other jurisdiction recognize animal sentience in their legislation” [78].

#### 4.1.3. Animal Speciesism?

Drawing on the philosopher Peter Singer’s original definition of ‘speciesism’ as “a prejudice or attitude of bias in favor of the interests of members of one’s own species and against those of members of other species” [267], speciesism more generally has been defined as “the unjustified disadvantageous consideration or treatment of certain individuals because they are not members of a given species” [268]. In relation to animal protection legislation, the variations in protection of particular animal species could potentially be argued to be a form of speciesism. Animal protection legislation is intended to ensure that animals are treated humanely [260] by prohibiting unnecessary, unjustifiable or unreasonable suffering [262]. However, as discussed by White [260], there is a point at which animal protection tends to cease when it is in conflict with human interests, and this point is vastly dependent on the context of animal use: consider for instance animals used as pets as compared to in the context of for-profit enterprises such as farming or when classified as pests. Thus, the legal protections awarded to animals typically are based on their utility to humans and their extrinsic value (worth to humans) rather than their sentient abilities or intrinsic or moral value [269]. Thus, the animals that we keep as pets are awarded far more protection in comparison to those we think of as pests or food-producing animals, regardless of their sentient abilities. Due to exemptions for compliance with delegated legislation and codes of practices, the level of suffering that a person can cause livestock, research or pest animals is much greater than that accepted in relation to animals in a companion context [261]. For example, tail docking can be performed on piglets without pain relief under the Pig Model Code of Practice for the Welfare of Animals [270]. However, if that same procedure were performed on a dog without pain relief, it would be classified as ‘unnecessary suffering’ and constitute an offence under the animal welfare acts. It has been claimed that laws are constructed to reflect the public’s concern [5], and sociological research has shown that the public have higher regard for companion animal welfare as compared to farm or feral animal welfare [269,271,272]: thus, perhaps it is unsurprising that the law does not protect all animals in the same way. However, these topics warrant much greater consideration and scrutiny, including about whether the applicability of various types of animal welfare legislation in Australia is unjustifiably limited to certain species of animals and how public opinion is evolving with regard to these issues.

### 4.2. What Is ‘Animal Welfare’?

#### 4.2.1. The Scientific Viewpoint

Within the scientific community there is no exact or universal definition of ‘animal welfare’ [273], but instead there are many different definitions and interpretations [274]. The main approaches to welfare focus on either the biological functioning of an animal where welfare equates to physiological stress responses and good physical health, or the animal’s affective state, where welfare focuses on how the animal feels [275] and its attempt to cope with its environment [276]. The World Organization for Animal Health (OIE) is an inter-governmental organization that has created an international definition of animal welfare, in an attempt to provide an accepted, baseline standard for countries to adhere to in their commercial relationships with animals. Countries can choose to become members and adopt the guidelines [277]. Australia, along with 182 other countries, is a member [278]. The OIE has defined animal welfare as “the physical and mental state of an animal in relation to the conditions in which it lives and dies” [277] and makes direct reference to the ‘Five Freedoms’ (refer to [279] for Freedoms). Similarly, at a more local level, it has been suggested that domestic animal welfare statutes are also underpinned by the Five Freedoms [280]. However, as Webster [281] notes, the Freedoms were never intended to represent the overall status of an animal’s mental state, and with the emerging scientific acknowledgement of animal sentience, animal’s mental or affective states have been the primary focus in defining welfare [282,283,284,285] This is apparent with the development and refinement of the ‘Five Domains’ model (see, e.g., [286]). Nowadays, animal welfare is generally defined as the minimization of negative experiences and affective states, whilst also promoting positive experiences and affective states [283,287,288]. In Mellor [284] words, “the overall objective is to provide opportunities for animals to ‘thrive’, not simply ‘survive’”, which is achievable through measures that go beyond merely meeting animals’ basic needs for food, water, husbandry requirements and disease prevention [283].

#### 4.2.2. Legislative Interpretation

A surprising finding of this review is that despite the almost universal naming of the state animal protection acts as ‘Animal Welfare Acts’, there was no definition of ‘animal welfare’ within any of those Acts. This absence is especially concerning given that one of the legislative objectives is to promote welfare [10,11,12,13,14,15,16,17]. It is questionable whether the acts can effectively promote a concept that they have not formally defined. However, it is challenging to translate the positive aspects of welfare into enforceable provisions that can help to achieve the legislative objective of promoting welfare, especially given that scientists do not yet have effective methods for assessing positive welfare across all species [289]. Instead of defining ‘welfare’, the acts have only defined ‘cruelty’, which is essentially any form of unnecessary, unreasonable or unjustifiable pain, harm or suffering experienced by an animal (Table 7). However, these terminologies are vague and difficult to interpret and have been criticized for weakening the scope of protection awarded to animals [290]. It has been stated in the US context that defining ‘unnecessary suffering’ is individualized for each person based upon variables such as personal experiences, cultural standards and spiritual beliefs [291]. Thus, despite two centuries of statutory language reforms and case law precedents, criminal justice systems throughout western countries still struggle to balance legal, academic, cultural and public opinions when it comes to defining animal cruelty [292]. It is therefore perhaps unsurprising that a progressive interpretation of ‘welfare’ to include positive aspects, has not yet made its way into statutes given the longstanding definition of ‘cruelty’ is still open to interpretation.

The current approach is to impose obligations on persons in charge of animals to provide them with their basic needs [260], including appropriate living conditions, nutrition, disease prevention and veterinary treatment [37]. Note that this approach differs from many descriptions of welfare by focusing solely on survival without any attention to thriving or other positive experiences [284]. Perhaps such a focus is not surprising, given that the legal process is mostly focused on punishment [293,294,295,296], and hence has limited leverage for promoting positive aspects of welfare. Hence, as in many other contexts, we can see that law is essentially the minimum standard for ensuring basic animal welfare, and not the gold standard for promoting higher levels of animal welfare. Perhaps this latter ideal is better achieved through voluntary audit or accreditation schemes which could be specifically tailored for each species and provide market incentives to conform.

### 4.3. Animal Welfare Provisions

#### 4.3.1. Protecting Welfare in Law

Laws are only as good as their enforcement, and the ability of animal protection statutes to provide punishment where welfare is threatened or diminished (and to a more limited extent to promote positive aspects of welfare) is dependent on how the legislative model defines welfare. The wording of the acts provides recognition of the ability for animals to be harmed or experience suffering as a result of either acts or omissions, where terms such as ‘harm’ and ‘suffering’ have statutory definitions. Those definitions often include forms of distress and injury to animals, and there is an evidential burden of proving in court that distress or injury has occurred [37]. In other words, animals must endure some degree of suffering before animal cruelty provisions can be applied and enforced. It has been argued that this phraseology directly contradicts the objectives of animal welfare acts, as prosecution only occurs to punish cruelty, and not prevent it [6]. This model hence regularly results in negative outcomes for animals [297]. The WA, TAS and VIC legislatures have gone some way in addressing this issue by including the words “likely to cause harm” [58,298], in addition to the more common usage of “harm” for some offences under their acts. This provides additional options for intervention before an animal has been subjected to any form of harm, and thus more adequately protects animal welfare. It is contended that this concept is worthy of consideration by the remaining Australian jurisdictions.

It should be noted that although not included in this review, animal welfare laws have further mechanisms of protection other than welfare provisions. Such an example is ‘animal welfare directions’ included in each jurisdiction’s animal welfare act [299,300,301,302,303,304,305,306]. These directions utilize a compliance strategy where appointed animal welfare inspectors can issue statutory directions to persons responsible to provide for an animal’s welfare [307]. This mechanism allows for the prevention of cruelty and suffering before it occurs, and failure to comply with these directions often results in the imposition of punitive sanctions, and possible seizure of the animals concerned [307]. This approach to protecting animal welfare is consistent with the objectives of the statutes cross-jurisdictionally.

However, laws also have an impact on offending by providing social deterrence. Such functions have been noted in the literature on drink driving statutes for drinking under the influence (DUI) offences [308,309,310,311], which are similar to many animal welfare offences in being summary offences [312]. Deterrence theory focuses on the ability of laws to deter members of society from committing illegal acts, as they believe that offenders will be caught and punished [311]. However, this type of deterrence works primarily for individuals who are least likely to commit offences: as research on DUI enforcement has indicated, individuals who are “extreme drink drivers” are not deterred by the possibility of punishment [308,309,310], as visceral and other factors influence the committing of crimes rather than intellect or rational decision-making [309]. Hence animal protection laws protect welfare through a two-pronged approach: (1) via deterrence which promotes good treatment of animals (because of knowledge that it is illegal to mistreat them), and (2) via punishment of those who act outside of legal norms [308,309,310,313].

#### 4.3.2. Whose Responsibility?

The primary responsibility for protecting animals from cruelty and maintaining their welfare lies with the ‘person in charge’ or the ‘owner’ of an animal. All animal welfare statues impose liability on ‘persons in charge’ of an animal in relation to animal welfare offences. An owner/person in charge under animal welfare statutes is not necessarily the person to whom the animal is registered or has proprietary interests. Instead, it is the person who has ‘custody and control’ [28,29,30,31,33,34,35,36] of the animal at the point of time of an offence. This definition is consistent throughout each jurisdiction (Table 4) and is flexible as it can be adapted to the particular circumstances of a case. Case law plays a particular role in this regard through informing statutory interpretation. The doctrine of precedent allows for the development and progression of law through case law, as judges must follow precedents made in higher courts [314]. An important precedent in this area is *RSPCA SA v Evitts*, where a dog handler employed by a security firm was found to be the ‘owner’ of the dog as the employee had ‘custody and control’ of the animal at the time it required veterinary attention [315]. Cases of this nature, where the boundaries of legal definitions can be tested, allow the law to progress without the need for statutory reform [314]. Thus, statutory harmonization is not required for the owner/person in charge definition.

#### 4.3.3. Offences against Animals

As documented in our review, there are a range of offences designed to protect animals from cruelty that are outlined under the animal welfare acts. These statutes commonly describe aggravated and basic cruelty offences, and duty of care breaches. The distinguishing features between these offences commonly hinge on the seriousness of consequences to the animal, and the intent behind the defendant’s actions [37]. For the prosecution, the evidential burden is reduced in establishing a strict liability offence of animal cruelty since they do not have to establish the subjective element of *mens rea*, or intent [37]. In modern law, the attendant circumstances around the offence often replace traditional *mens rea*, indicating the level of culpability. Aggravated offences recognize that some acts are carried out with a greater degree of culpability and the offender therefore poses a greater danger to society. As a result, a greater penalty is prescribed [316]. However, not all statutes include an aggravated offence, which makes them non-uniform based on the research question. In spite of this, aggravating factors contributing to culpability, such as lack of remorse by the defendant or acting with others, are outlined under the states and territories sentencing statutes [112,317,318,319,320,321]. However, the inclusion of aggravation in animal welfare statutes acknowledges that not all cruelty is the same, and that acts causing high degrees of animal suffering should result in an increased punitive response. This can also be said for those animal welfare offences included in crimes statutes (Table 10). Crimes statutes mainly include offences against animals that go against human morals, such as bestiality [322]. Consequently, these offences attract the highest maximum penalties in comparison to all other protection provisions reviewed in this paper, implying that crimes statutes include the most serious offences involving animals with the highest level of culpability.

Animal protection provisions in wildlife statutes are generally consistent, as they all pertain to the prevention of injuring, harassing or killing wildlife (Table 11). There are different maximum penalties dependent on the status of a certain species (e.g., endangered, critical or vulnerable) [140,141,142,143,144,145,146,147,148,149]. The severity of the outcome resulting from injuring or killing an endangered species is much higher due to the harm not only to the individual animal but also the risk of or contribution to genetic loss, and consequently attracts a higher maximum penalty [323]. However, this implies that these ‘protected species’ are viewed primarily from a conservation perspective in that their genetics have value, rather than a welfare perspective acknowledging their sentient abilities. Further evidence in support of this is that nature conservation often fails to distinguish between fauna (animals) and flora (plants) [37], which overlooks the central distinction between the two groups, animals are sentient. This makes for a legal regime that pays little regard to wild animals’ sentient abilities [37], instead focusing primarily on their genetic diversity and value to conservation.

#### 4.3.4. Penalties and Punishment

Animal welfare offences and their corresponding maximum penalties have been the subject of numerous amendments cross-jurisdictionally, with reforms occurring over the last 20 years in QLD [324], SA [325], Victoria [326] and most recently NT [327]. The maximum penalties have been argued to represent parliamentary intent with regards to animal welfare [6], as they provide a benchmark against which the gravity of an offence should be measured [328] and are reserved for the worst, most serious examples of an offence [329,330]. It has been claimed that these amendments have signaled the intention of parliaments to “get tough” on animal welfare offenders, by sending a message that animal cruelty will not be tolerated [331,332]. Although the maximum penalties for offences are significant (Table 5), less than 10% of the maximum penalty is typically used in court proceedings [333], which raises questions about the role of statutory maximums. Hence it could be argued that maximum penalties are primarily intended as a symbolic gesture to show society the seriousness of the offence rather than being actual tools to improve animal welfare.

Promotion of animal welfare in the acts is primarily achieved by discouraging duty of care breaches through imposition of obligations on persons in charge of animals to provide for the basic needs of animals [260]. However, when deterrence fails, intervention and punishment is required [309,310]. Deterrence is, however, only one facet of punishment theory, with the other facets being rehabilitation, retribution, restitution and incapacitation [293,294,295,296]. Previous research into other summary offences has suggested that without legal intervention, including punishment, offenders will continue to push the limits of laws [310]. However, it has been suggested that the commonly imposed forms of penalty, namely imprisonment and fines, are not the most effective ways to punish animal welfare offenders. Consideration should instead be given to alternative and more rehabilitative forms of penalties [6,313,333]. Such a conclusion was found from Ghasemi [309] research into criminal penalties, where it was suggested that legal systems should consider the fundamentals of criminal acts and attempt to reduce them with strategies other than punishment. It was noted that punishment may not “solve the problem” for criminal offenders [309] and may be ineffective in reducing recidivism. Alternative penalties for animal welfare offences include court mandated counselling and non-violent-conflict resolution training [313]. These mechanisms are more likely to tackle the root of the problem and actually help the offender [333], which would in turn reduce the likelihood of reoffending. Consideration of such penalties is especially important given the established link between animal abuse and human violence [334,335,336,337,338,339,340,341].

#### 4.3.5. Court Orders

Court orders are directions issued by courts to an offender found guilty of an offence (prohibition or supervision orders) or alleged offenders undergoing investigation (interim orders), the provisions of which are outlined in the state and territory acts [80,81,82,83,84,85,86,87]. Enforcement authorities commonly apply to the court for orders as a mechanism to further protect animals [342]. The main issue with these court orders is that they are not cross-jurisdictionally recognized in most states and territories, exceptions being NSW [92], Tas [93] and VIC [94]. This lack of interstate recognition allows offenders who have been prosecuted and placed under a court order in one jurisdiction to cross the border and be free of the provisions therein [297] (see, e.g., [343]). Recognition of interstate orders could provide tangible welfare benefits by preventing known offenders from having animals, and is an area worth considering for legal reform [297]. Currently in Australia, recognition of interstate orders has been achieved for other types of offending, such as in the case of domestic violence orders [344]. Given this, as well as the three of the eight Australian jurisdictions already implementing such provisions into their animal protection legislation (Table 8), there is legal precedent for making such a change in relation to animal welfare court orders.

There are three jurisdictions that have extra provisions attached to their court orders. Generally, court orders can only be handed down once an offender has been either convicted or found guilty of an offence [80,81,82,83,84,85,87]. However, Section 12(1) of the *Prevention of Cruelty to Animal Act 1986 (VIC)* [345] and Section 32A(1) of the *Animal Welfare Act 1985 (SA)* [346] allows court orders to be issued to persons found not guilty due to mental impairment, which is an important provision particularly in relation to supervising and restricting animal hoarders. Animal hoarding is a mental illness [347,348], and thus prosecuting and penalizing people who suffer from that illness may not be the most effective intervention for either the offender or the animals. Prosecutions are lengthy and animals are often kept in shelters whilst the case is underway [349], for evidentiary purposes [350]. Thus, if prohibition orders can be given for persons hoarding animals without a guilty finding, it would prevent animals being held in a legal limbo, and the persons being punished with a penalty that may not deter them from reoffending in the future. Another provision from Section 104A(3) of the *Animal Welfare Act 1992 (ACT)* allows the courts to issue a mandated counselling order [351], which provides a more rehabilitative strategy that may ‘better fit the crime’ [6,313,333]. Although it is not possible to assess the enforceability of these mechanisms, these provisions in the ACT, SA and VIC animal welfare statutes appear to be attempting to utilize alternative forms of penalties in special cases, and other jurisdictions in Australia would benefit from considering these options.

### 4.4. Avenues for Reform

#### 4.4.1. Harmonizing Laws

In spite of Constitutional restrictions, the harmonization of animal welfare statutes could be achieved through the creation of Uniform Acts adopted independently by each state and territory. Uniform Acts are prepared and adopted voluntarily in whole or substantially by each state and territory [352]. An example of such an approach in Australia is the Uniform Evidence Acts, which have been adopted by NSW [353], VIC [354], TAS [355], ACT [356], NT [357]; Parliaments in SA, QLD and WA are yet to follow. Although a uniform approach to state and territory-based legislation would appear in theory to be beneficial, it still comes with its challenges. Firstly, the harmonization of the evidence acts began in 1995, and two decades later all states are yet to comply [358], indicating that a uniform approach is a slow process and that beneficial results may take years to be realized. Secondly, the application of this uniform approach has been suggested to be more dis-uniform in practice [358] than first might be thought. Priest [358] has discussed numerous occasions extending over a decade of differences in application of the Uniform Evidence Acts across states. Thus, even if a uniform approach were to be adopted for animal protection statues, there is no guarantee that it would be applied similarly and consistently across jurisdictions, given the potential differences between enforcement agencies across the states.

#### 4.4.2. Use of Delegated Legislation

One thing noted from this review was the lack of consistency between welfare provisions included in, as well as the legislative subject domains of, the miscellaneous acts (Table 13). For example, the QLD *Exhibited Animals Act 2015* [202] was included in this review as it allows inspectors to seize any exhibited animal for welfare reasons. However, the NSW *Exhibited Animals Protection Act 1986* [359] was excluded as it included no welfare provisions and is largely administrative based. It is surprising that two acts within the same subject domain, lack this degree of uniformity when it comes to animal welfare. This same trend was observed for the majority of the included miscellaneous statutes and has since been acknowledged as an issue in the Victorian Government’s proposal for animal welfare reforms. It was stated that “while the POCTA Act is currently the primary legislation for managing animal welfare in Victoria, the Act does not apply in all situations where an animal is being used, handled or managed. This is one of the more confusing aspects for the community and can create challenges for regulators and those who are trying to comply with the rules” [78]. Without a clear map or explanation depicting the relationship between the miscellaneous statutes at the state-level, it is hard to critique their effectiveness in maintaining animal welfare. This issue mostly arises due to the use of delegated legislation, where animal welfare provisions are often included in the statute’s corresponding regulations instead of the statute itself. Parliament will lay down principles of new laws through statutes, whilst delegated legislation will establish the application of those new laws in greater detail, making it flexible in matters subject to frequent changes [360]. However, in the animal law context, the inconsistent use, and in some cases the potential overuse, of delegated legislation can lead to uncertainty around the importance of animal welfare as a policy consideration, given that delegated legislation lacks parliamentary oversight [361]. Without a comprehensive assessment of delegated legislation, it is difficult to say the extent to which it contributes to dis-uniformity between the states and territories. A potential avenue for reform could involve bringing important provisions currently placed in the delegated legislation, into the enabling statute, to increase their weight. Exclusion of delegated legislation in the forms of regulations and codes of practice is a limitation to this study, as statutes do not work independently [220]. In Australia, as a common law country, statutes, regulations, codes of practice and the doctrine of precedent work symbiotically to form a coherent corpus of law [220]. Therefore, it should be noted that a further review of these excluded components is required to form a comprehensive understanding of the shortcomings of animal protection legislation, and that this review alone only examines a single facet of animal protection law.

#### 4.4.3. National Oversight

This review was limited to black letter law. The actual outcomes resulting from law are influenced by a myriad of factors, including enforcement and resourcing, and thus there may be better ways to achieve a more coherent approach to animal welfare law rather than statutory harmonization. One strategy would be to appoint a single body to represent and oversee animal protection legislation. Such an approach has previously been attempted in Australia. The Australian Animal Welfare Strategy (AAWS), led by the Federal Government, was developed to establish a national approach to standard setting in an attempt to bring the states and territories together [3,260]. The AAWS aimed to improve coordination and deliver a more effective approach to improving animal welfare [362]. However, funding for the AAWS ceased as part of the Australian Liberal Party’s 2014-2015 budget measures [3], and there has been little progress in furthering the AAWS’s aims since. There have been ongoing discussions about the need for national oversight for animal welfare. For instance, when the Greens party attempted to introduce a bill for the establishment of an independent authority in 2015, and the Australian Labour Party made a promise to create such an office in 2016. The Coalition, however, elected not to support the national body due to cost concerns [363]. In addition, there have been numerous independent reviews and inquiries into animal welfare enforcement in both Australia [364,365,366] and internationally [367,368], with the Canadian province of Ontario changing their enforcement model as a result [369], and recent proposals have been made for New Zealand to establish an independent authority to oversee animal welfare at a national level [368]. However, in Australia, the conclusions drawn from the two reviews conducted in Victoria [364] and WA [365] was that the current enforcement models are appropriate and an independent authority is not required (see [6] for further commentary). Nevertheless, these discussions and reviews show that Parliamentarians and policy makers continue to consider a national independent body for animal welfare in Australia. It could be likely in the future that Australia will follow the emerging international trends of developing such a body, given the potential benefits that it would provide such as capabilities for national data collection [3] and enforcement training, consolidation of resourcing and more control over conflicts of interest including policy being driven by industries [370]. Such an approach would likely be more effective in improving animal law enforcement than statutory harmonization.

## 5. Conclusions

Animal protection provisions in Australian law are more extensive than many might think. Animal welfare acts clearly make major contributions to animal protection from a legal standpoint, but other acts play roles in more targeted manners through their focus on particular species or classes of animals. Animal protection laws are broadly consistent throughout Australian jurisdictions but were found in this analysis to have some shortcomings, including lack of clear and coherent definitions of ‘animal’ and other key terms. The enforcement mechanisms associated with animal welfare warrant much more attention, given the questionable effectiveness of using legal punishment to promote welfare and prevent cruelty; greater consideration should be given to alternative forms of penalties particularly rehabilitation. Theoretically, a generally uniform approach to animal protection would be beneficial. However, given the Constitutional restrictions in Australia, uniformity it is not likely to be feasible, especially without the addition of a national oversight scheme or body. In the absence of such a body, a relatively efficient approach would be for jurisdictions to take note of reform efforts made by other jurisdictions, such the ACT’s recent recognition of animal sentience or the interstate prohibition orders in NSW, TAS and VIC. Finally, as this review was only limited to statutory written law, in-depth research is required on delegated legislation as well as enforcement statistics in order to fully assess the effectiveness of current legal regimes for animal welfare, as laws are only as good as their enforcement.

## Figures and Tables

**Figure 1 animals-11-00035-f001:**
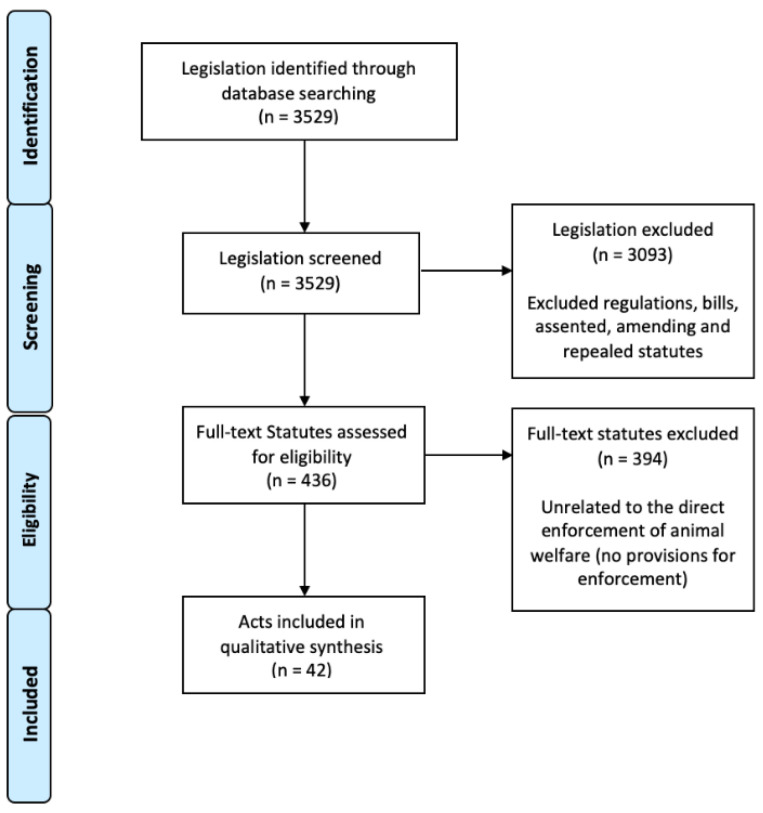
PRISMA [9] flow diagram showing the data selection process of animal protection statutes.

**Table 1 animals-11-00035-t001:** Statutes which include provisions for animal welfare protection in various Australian jurisdictions. ‘Other’ categories include chemical use statutes, racing statutes, a summary offences statute and a police powers statute. For a detailed list of all 42 statutes, refer to Appendix A.

Jurisdiction	Animal Welfare	Crimes	Wildlife/ Environment	Fish	Livestock	Sports	Zoo	Other	Total
Commonwealth (CTH)			✓✓						2
Australian Capital Territory (ACT)	✓	✓	✓	✓					4
New South Wales (NSW)	✓	✓	✓✓	✓				✓	6
Northern Territory (NT)	✓	✓	✓	✓				✓	5
Queensland (QLD)	✓	✓	✓			✓	✓	✓	6
South Australia (SA)	✓	✓	✓	✓				✓	5
Tasmania (TAS)	✓	✓	✓	✓	✓				5
Victoria (VIC)	✓	✓	✓	✓	✓	✓			6
Western Australia (WA)	✓	✓	✓						3

**Table 2 animals-11-00035-t002:** Animal welfare statutes for each Australian state and territory.

Jurisdiction	Statute
ACT	*Animal Welfare Act 1992* [18]
NSW	*Prevention of Cruelty to Animals Act 1979* [19]
NT	*Animal Welfare Act 1999* [20]
QLD	*Animal Care and Protection Act 2001* [21]
SA	*Animal Welfare Act 1985* [22]
TAS	*Animal Welfare Act 1993* [23]
VIC	*Prevention of Cruelty to Animals Act 1986* [24]
WA	*Animal Welfare Act 2002* [25]

**Table 3 animals-11-00035-t003:** Statutory definition of ‘animal’ in each state and territories’ animal welfare act. References as follows: ACT [28], NSW [29], NT [30], QLD [27], SA [31], TAS [32], VIC [26], and WA [33].

Jurisdiction	Mammals	Reptile	Amphibian	Birds	Fish	Crustacean	Cephalopod
ACT	✓	✓	✓	✓	✓	✓ *	✓
NSW	✓	✓	✓	✓	✓	✓ *	
NT	✓	✓	✓	✓	✓ **	✓	
QLD	✓	✓	✓	✓	✓	✓	✓
SA	✓	✓	✓	✓			
TAS	✓	✓	✓	✓	✓		
VIC	✓	✓	✓	✓	✓	✓	
WA	✓	✓	✓	✓			

* Only if used for human consumption. ** Captive fish only.

**Table 4 animals-11-00035-t004:** Statutory definition of ‘owner/person in charge’ in each state and territories’ animal welfare Act.

Jurisdiction	Owner/Person in Charge
ACT	“custody or control” [28].
NSW	“possession or custody” or “care, control or supervision” [29].
NT	“possession” where possession includes custody, care, control or supervision [30].
QLD	“custody” where custody includes care and control, or “proprietary interests” [34].
SA	“custody and control” [31].
TAS	“control, possession or custody” [35].
VIC	“possession or custody” or “care, control or supervision” [36].
WA	“actual physical custody or control” [33].

**Table 5 animals-11-00035-t005:** Each animal welfare offence and the corresponding maximum penalties per each state and territory’s animal welfare act. Penalties are applicable to natural persons; separate penalties apply for body corporates.

Jurisdiction	Duty of Care Breach	Basic Cruelty	Aggravated Cruelty
ACT	Sections 6B-G [43]	Section 7 [44]	Section 7A(1) and (2) [45]
	Monetary: varies between 25 and 100 penalty units	Monetary: 200 penalty units	Monetary: 300 penalty units
	Custodial: 1 year (s6B and 6G)	Custodial: 2 years	Custodial: 3 years
NSW	Sections 8–11 [46]	Section 5 [47]	Section 6(1) [39]
	Monetary: 50 penalty units	Monetary: 50 penalty units	Monetary: 200 penalty units
	Custodial: 6 months	Custodial: 6 months	Custodial: 2 years
NT	Section 8(2) [48]	Section 9(1) [49]	Section 10(1) [50]
	Monetary: 100 penalty units	Monetary: 150 penalty units	Monetary: 200 penalty units
	Custodial: 1 year	Custodial: 18 months	Custodial: 2 years
QLD	Section 17(2) [51]	Section 18(1) [52]	Not included
	Monetary: 300 penalty units	Monetary: 2000 penalty units	
	Custodial: 1 year	Custodial: 3 years	
SA	Included in Section 13(2) under definition of s13(3)(b)	Section 13(2) [53]	Section 13(1) [54]
		Monetary: $20,000.00	Monetary: $50,000.00
		Custodial: 2 years	Custodial: 5 years
TAS	Section 6 [55]	Section 8(1) [56]	Section 9(1) [57]
	No penalties included	Monetary: 100 penalty units	Monetary: 200 penalty units
		Custodial: 1 year	Custodial: 5 years
VIC	Included in Section 9(1)	Section 9(1) [58]	Section 10(1) [38]
		Monetary: 250 penalty units	Monetary: 500 penalty units
		Custodial: 1 year	Custodial: 2 years
WA	Included in Section 19(1) under definition of s19(3)	Section 19(1) [59]	Not included
		Monetary: $50,000	
		Custodial: 5 years	

**Table 6 animals-11-00035-t006:** Each offence pertaining to participating in prohibited activities and using prohibited items on animals, and the corresponding maximum penalties per each state and territory’s animal welfare act. Penalties are applicable to natural persons; separate penalties apply for body corporates. Note: In a number of jurisdictions, there is referral to associated regulations for specific conditions, which are not listed here.

Jurisdiction	Provisions (Activities)	Activities Prohibited	Provisions (Items)	Items Prohibited
ACT	Sections 17–18A [60] Monetary: 300 penalty units Custodial: 3 years	Violent animal activities Rodeos and game parks Greyhound racing	Sections 13–14 [61] Monetary: 100 units Custodial: 1 year	Electrical devices
NSW	Sections 18–21C * [62] Monetary: 50 penalty units Custodial: 6 months	Animal baiting and fighting Trap shooting Game parks Certain animal catching activities Coursing Firing Tail nicking Steeple chasing and hurdling	Sections 16–17 [63] Monetary: 50 penalty units Custodial: 6 months	Electrical devices Certain spurs
NT	Section 21 [64] No penalties included	Animal competitions Release of animal for purpose of hunting Animal fighting and baiting	Sections 18–20 [65] Monetary: 10 penalty units Custodial: N/A	Traps Electrical devices Spurs
QLD	Section 21(1) [66] Monetary: 300 penalty units Custodial: 1 year	Animal fighting Hunting (under certain conditions) Coursing	Section 35 [67] Monetary: 300 penalty units Custodial: 1 year	Trap or spur as prescribed in Regulations
SA	Section 14(1) [68] Monetary: $50,000 Custodial: 4 years	Animal fighting Live baiting Release of animal for purpose of hunting	Section 14A(1) [69] Monetary: $20,000.00 Custodial: 2 years	Cock-fighting spur Lures or bait Items used in animal fighting
TAS	Section 10(1) [70] Monetary: 200 penalty units Custodial: 1 year	Animal fighting Release of animal for purpose of hunting Live baiting	Section 12(1) [71] Monetary: 100 penalty units Custodial: 1 year	Certain traps
VIC	Sections 13–14 [72] Monetary: 500 penalty units Custodial: 2 years	Baiting and luring Animal fighting Trap shooting Inciting dog to chase other animal	Section 15AB(1) [73] Monetary: 240 penalty units Custodial: 2 years	Certain traps
WA	Section 32(1) [74] Monetary: $50,000 Custodial: 5 years	Release of animal for purpose of hunting Animal fighting	Section 31(1) [75] Monetary: $20,000 Custodial: 1 year	Things intended to inflict cruelty

* Section 20 of NSW Prevention of Cruelty to Animals Act 1979 has a maximum penalty of 200 penalty units or 2 years imprisonment, or both.

**Table 7 animals-11-00035-t007:** Definition of cruelty in the corresponding state and territory animal welfare statute.

Jurisdiction	Definition
ACT	doing, or not doing, something to an animal that causes, or is likely to cause, injury, pain, stress or death to the animal that is unjustifiable, unnecessary or unreasonable in the circumstances [76].
NSW	any act or omission as a consequence of which the animal is unreasonably, unnecessarily or unjustifiably inflicted with pain [29]. *
NT	causes the animal unnecessary suffering [49]. *
QLD	causes [the animal] pain that, in the circumstances, is unjustifiable, unnecessary or unreasonable [52]. *
SA **	intentionally, unreasonably or recklessly causes the animal unnecessary harm [79]. *
TAS	any act, or omit to do any duty, which causes or is likely to cause unreasonable and unjustifiable pain or suffering to an animal [56].
VIC	No definition included. ***
WA	in any way causes the animal unnecessary harm [59]. *

* Other specific definitions are included. ** SA uses the term ‘ill treatment’ instead of the commonplace ‘cruelty’ used throughout the remaining jurisdictions. *** VIC uses specific examples of what constitutes ‘cruelty’ under Section 9, but there is no strict definition [77].

**Table 8 animals-11-00035-t008:** Different types of court issued orders outlined in each states and territories’ animal welfare Acts.

Jurisdiction	Prohibition	Supervision	Interim	Interstate
ACT	Section 101A [80]		Section 100A [90]	
NSW	Section 31 [81]			Section 31AA [92]
NT	Section 76A [82]			
QLD	Section 183 [83]		Section 181A [91]	
SA	Section 32A(1)(d) [84]	Section 32A(1)(aa) [88]		
TAS	Section 43 [85]			Section 43AAB [93]
VIC	Section 12(1)(a) [86]	Section 12(1)(b) [89]		Section 12A [94]
WA	Section 55 [87]			

**Table 9 animals-11-00035-t009:** Crime statutes per each Australian state and territory.

Jurisdiction	Statute
ACT	*Crimes Act 1900* [103]
NSW	*Crimes Act 1900* [104]
NT	*Criminal Code Act 1983* [105]
QLD	*Criminal Code Act 1899* [106]
SA	*Criminal Law Consolidation Act 1935* [107]
TAS	*Criminal Code Act 1924* [108]
VIC	*Crimes Act 1958* [109]
WA	*Criminal Code Act Compilation Act 1913* [110]

**Table 10 animals-11-00035-t010:** Each offence relating to animal protection included in the Australian state and territories’ crimes acts. Offences are broken down into acts of aggravated animal cruelty, bestiality and provisions to protect working animals. Offences have been paraphrased. Penalties are applicable to natural persons; separate penalties apply for body corporates.

Jurisdiction	Aggravated Cruelty	Bestiality	Working Animals *
ACT		Section 63A [117]—offence to commit bestiality. Max penalty: 10 years imprisonment.	
NSW	Section 530 [114]—offence to intend to (s530(1)) or be reckless about (s530(1A)) harming an animal. Max penalty: (1) 5 years and (1A) 3 years imprisonment.	Section 79 [118]—offence to commit bestiality. Max penalty: 14 years imprisonment. Section 80—offence to attempt to commit bestiality. Max penalty: 5 years imprisonment.	Section 531(1) [115]—offence to intentionally kill or seriously injure an animal used for law enforcement. Max penalty: 5 years imprisonment.
NT		Section 138 [119]—offence to commit bestiality. Max penalty: 3 years imprisonment.	
QLD	Section 242(1) [120]—offence to intend to kill or inflict severe pain on an animal. Max penalty: 7 years imprisonment. Section 468(1) [113]—offence to intentionally wound or kill an animal. Max penalty: 7 years imprisonment if committed on stock animals. Max penalty: 2 years for any other animals or 3 years imprisonment if committed at night.	Section 211 [121]—offence to have carnal knowledge with or of an animal. Max penalty: 7 years imprisonment.	
SA		Section 69 [122]—offence to commit bestiality. Max penalty: 10 years imprisonment.	Section 83I(1) [123]—offence to intentionally cause death or serious harm to a working animal. * Max penalty: 5 years imprisonment.
TAS		Section 122 [124]—offence to engage in an act of bestiality. Max penalty: not included.	
VIC		Section 54A(1) [125]—offence to commit bestiality. Max penalty: 5 years imprisonment.	
WA		Section 181 [126]—offence to have carnal knowledge of an animal. Max penalty: 7 years imprisonment.	

* Definition of working animals includes animals used for law enforcement (NSW) [115] and police dogs, police horse, correctional services dog or accredited assistance dogs (SA) [116].

**Table 11 animals-11-00035-t011:** Each wildlife protection statute and the corresponding animal protection provisions per each Australian state and territory. Statutes only award protection to native wildlife species in Australia. Offences have been paraphrased.

Jurisdiction	Statute	Animal Protection
CTH	*Environment Protection and Biodiversity Conservation Act 1999* [150]	Section 196(1) [141]—offence to kill or injure threatened species. Section 211(1) [151]—offence to kill or injure migratory species. Section 229(1) [152]—offence to kill or injure a cetacean. Section 254(1) [153]—offence to kill or injure listed marine species. Section 303GP(1) [154]—offence to treat export or import animals in a cruel manner.
	*Great Barrier Reef Marine Park Act 1975* [155]	Section 38GA(1)(c)(i) [156]—offence to take or injure a protected animal species.
ACT	*Nature Conservation Act 2014* [157]	Section 130(1) [140]—offence to kill a native animal. Section 131(1) and (2) [158]—offence to (1) injure or (2) endanger native animals.
NSW	*National Parks and Wildlife Act 1974* [159]	Section 45(1)(a) [160]—cannot harm any animal within a national park. Section 56(1) [161]—cannot harm any animal within a nature reserve. Section 70(1) [162]—cannot harm any fauna within a wildlife refuge, conservation area or wilderness area.
	*Biodiversity Conservation Act 2016* [163]	Section 2.1(1) [142]—offence to harm threatened or protected species. Section 11.32 [164]—animal welfare directions to people who keep protected animals in confinement. Section 11.36 [165]—offence to contravene animal protection direction.
NT	*Territory Parks and Wildlife Conservation Act 1976* [166]	Section 66(1) [143]—offence to interfere with protected wildlife (where interfere with includes harm or disturb). Section 67 [167]—offence to interfere with unprotected wildlife.
QLD	*Nature Conservation Act 1992* [168]	Section 88(a) [144]—cannot take protected animals (where take includes injure or disturb).
SA	*National Parks and Wildlife Act 1972* [169]	Section 45(1) [145]—cannot take an animal within a sanctuary. Section 51(1) [170]—cannot take a protected animal (where take includes injuring). Section 65(1) [171]—cannot poison animals with the intent of taking them. Section 68(1)(a) [146]—cannot harass or molest protected animals.
TAS	*Threatened Species Protection Act 1995* [172]	Section 51(a) [173]—offence to take fauna (where take includes injure).
VIC	*Wildlife Act 1975* [174]	Section 21(3) [175]—cannot take, destroy, disturb or injure wildlife. Section 54(1) [176]—cannot poison animals with the intent to injure them. Section 58(a) and (b) [177]—cannot (a) molest or injure, or (b) disturb protected wildlife. Section 76(1)(a) [178]—cannot injure, kill or take whales.
WA	*Biodiversity Conservation Act 2016* [179]	Section 149(1) [147]—offence to take fauna (where take includes injuring). Section 150(1) [148]—offence to take threatened fauna. Section 153(1) [149]—offence to disturb fauna.

**Table 12 animals-11-00035-t012:** Fish statutes and the corresponding animal protection provisions per each Australian state and territory excluding QLD and WA, as there is no animal protection awarded in those statutes. Offences have been paraphrased for brevity.

Jurisdiction	Statute	Animal Protection
ACT	*Fisheries Act 2000* [186]	Section 88A(1)(a) [187]—offence to disturb spawning fish in public waters.
NSW	*Fisheries Management Act 1994* [188]	Section 207(2) [189]—offence to disturb spawning fish.
NT	*Fisheries Act 1988* [190]	Section 11(4) [191]—offence to intentionally use shock systems that detrimentally affects fish life.
SA	*Fisheries Management Act 2007* [192]	Section 71(1)(b) and (2)(a) [193]—offence to injure (1)(b) or harass (2)(a) protected aquatic mammals or fish.
TAS	*Living Marine Resources Management Act 1995* [194]	Section 255(1)(c) [195]—offence to use shock systems that are likely to injure or detrimentally affect fish life.
VIC	*Fisheries Act 1995* [196]	Section 71(1) [197]—offence to injure protected aquatic biota without permit.

**Table 13 animals-11-00035-t013:** Miscellaneous statutes with inclusions of animal protection provisions. These statutes can be further categorized into livestock, racing/sports, exhibited animals, chemical use, police powers and summary offences. ACT and WA are not included as no additional statutes that fit into this category were not sourced. Offences have been paraphrased for brevity.

Jurisdiction	Statute	Animal Protection
NSW	*Radiation Control Act 1990* [198]	Section 24(1) [199]—if the perpetrator of any offence committed under the Act was reckless about causing serious harm to a human, animal or the environment then the maximum penalty can be increased to 1500 units or 2 years imprisonment, or both.
NT	*Agricultural and Veterinary Chemicals (Control of Use) Act 2004* [200]	Section 13(b) [201]—A person who uses a chemical product, fertilizer or stock food must take all measures that are reasonable and practicable to ensure the use does not result in harm to humans, animals or the environment.
QLD	*Exhibited Animals Act 2015* [202]	Section 192(2)(d)(i) and (ii) [203]—The inspector may seize an exhibited animal or other thing at the place if the inspector reasonably believes the interests of the welfare of the animal require its immediate seizure.
	*Police Powers and Responsibilities Act 2000* [204]	Section 143 [205]—police officers can give animal welfare directions. Section 146 [206] and 147 [207]—outlines powers given to police officers to alleviate any suffering experienced by animals. Section 689(2)(a) [208]—animals in the police service must be kept in a way that maintains the animals’ welfare and the welfare of other animals.
	*Racing Integrity Act 2016* [209]	Section 193(1) [210]—authorized officers under the Act may give animal welfare directions. Section 218(1)(a) and (b) [211]—offence to use (a) prohibited items or (b) interfere with licensed animals (where “interfere with” includes injure or detrimentally affect the licensed animals and “licensed animal” means any animal that is licensed for racing purposes or undergoing a trial to become a licensed animal at a licensed venue).
SA	*Summary Offences Act 1953* [212]	Section 17C(1) [213]—A person who, while trespassing on land on which animals are kept in the course of primary production, disturbs any animal and thus causes harm to the animal, or loss or inconvenience to the owner of the animals, is guilty of an offence.
TAS	*Animal Farming (Registration) Act 1994* [214]	Section 24(1) [215]—A person must not unlawfully kill or injure a prescribed animal that is being farmed in accordance with the Act, where prescribed animals are outlined in schedule 1 in the corresponding regulations (emu and sheep species).
VIC	*Livestock Management Act 2010* [216]	Section 50(1)(b) [217]—offence to act or fail to act in a manner that results in serious risk to animal welfare.
	*Racing Act 1958* [218]	Section 55(1) [219]—offence to be present at a greyhound race where an animal is being used as a lure.

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
