# Peer review of "Assessing the Uniformity in Australian Animal Protection Law: A Statutory Comparison"

_animals, 2020, doi:10.3390/ani11010035_

Round 1

Reviewer 1 Report

l 131Table 1 - number of ticks does not seem to relate to the Total at the end

l. 217/l. 342 be interested to see how old each Act is and how much they relied on previous UK legislation as attitudes to animals has changed so much particularly in last 20 years eg ACT legislation is 2019 and so has sentience included - a table showing when the framework legislation was last updated would be helpful eg it seems WA followed by Tasmania have the oldest legislation and ACT and Victoria the most recent - has this made a difference to how animal welfare is maintained in each State or even state to animal cruelty (as measured by per capita convictions, providing enforcement is the same 

l. 378 inclusion of cephalopods probably is a function of the age of the legislation - cephalopods have only been recognised as sentient in last 15 years

l 470 would be worth giving a reference to the OIE's definition of animal welfare as this is the global body for animal welfare and Australia is a member - if a State is falling below the OIE definition of animal welfare then it is an interesting point as to whether the Federal Government, as the OIE member, should ask the State to update their definition

l 499 many pieces of more up to date legislation on animal welfare refer to the 5 animal needs (what the animal needs) or domains (what the animal has in a positive state) (eg AWA 2006 UK) and you have to assess the welfare of the animal against these to assess if those needs are being met 

l 610 I think the idea of sentencing being a deterrent has yet to be proved but this assertion needs references if any are available - I doubt anyone would think twice about being cruel to an animal because of the sentence unless it was a business decision eg illegal puppy farming, transport of animals were it may be a deterrent

l 629 is there any data on recidivism in the different states relating to animal offence or the level of penalty that the State has eg how many times has the maximum sentence been applied in each State

l. 686 is there any evidence that either legislation (QLD or NSW) has had more impact on animal exhibits eg they tend to be based on one state as the rules are more law there or dont travel to a state due to the laws there

l 724 it would be useful to have data from each State on convictions per capita to show impact or otherwise of the different legislations on the animal cruelty offences eg is there less cruelty per capita in a State with a more modern approach to how the law is written  or with mens rea included or not 

Author Response

  1. 131Table 1 - number of ticks does not seem to relate to the Total at the end

We have included the correct number of ticks to reflect the specific numbers of statutes in each category. We have also included in the table legend a referral to the appendix for the full list of included statutes

217/l. 342 be interested to see how old each Act is and how much they relied on previous UK legislation as attitudes to animals has changed so much particularly in last 20 years eg ACT legislation is 2019 and so has sentience included – a table showing when the framework legislation was last updated would be helpful eg it seems WA followed by Tasmania have the oldest legislation and ACT and Victoria the most recent – has this made a difference to how animal welfare is maintained in each State or even state to animal cruelty (as measured by per capita convictions, providing enforcement is the same 

Thank you for this suggestion. We are extremely interested in this particular topic of reviewing the recent animal law reforms in each Australian state and territory. However, we believe it would be most valuable to review the entirety of the history dating back to the commencement of the oldest legislation (POCTA Act 1979 NSW) and trace all the amendments to assess if there are any trends within the jurisdictions. This is because when we looked at this data, we found that the most recent major amendments are not very valuable on their own as many reforms only focus on specific sections of the Act (eg, ACT in 2019 added sentience, but NSW in 2018 added regulations for trading animals, and in 2018 WA simply changed their object clause). This data on its own does not tell you much of the story, but the entire history could outline trends. It would require much data collation and discussion around societal trends and the parliamentary second reading speeches, which in this manuscript we wouldn’t be able to dedicate the word count too. We sincerely appreciate this comment and we do plan to analyse the history of animal law reforms in Australia, but we believe it would be best suited to its own manuscript.

378 inclusion of cephalopods probably is a function of the age of the legislation - cephalopods have only been recognised as sentient in last 15 years

This is very interesting, and we did check when cephalopods were included into the ACT and QLD legislation – ACT amended cephalopods into the AWA in 2000, and QLD’s AWA was created in 2001 with cephalopods already included (no amendments). Again, as above, we would love to discuss this in detail when reviewing the history to animal law reforms, as we would be able to analyse parliamentary debates on this topic and uncover the reasoning behind their inclusion. We just do not believe we could give it the in-depth discussion it deserves in this manuscript.

l 470 would be worth giving a reference to the OIE's definition of animal welfare as this is the global body for animal welfare and Australia is a member - if a State is falling below the OIE definition of animal welfare then it is an interesting point as to whether the Federal Government, as the OIE member, should ask the State to update their definition

Thank you for this recommendation. We have included it on lines 485-489.

l 499 many pieces of more up to date legislation on animal welfare refer to the 5 animal needs (what the animal needs) or domains (what the animal has in a positive state) (eg AWA 2006 UK) and you have to assess the welfare of the animal against these to assess if those needs are being met 

Australian animal welfare legislation is yet to recognise Mellor’s Five Domains model, as the five freedoms are still regularly discussed in parliamentary debate documents, with ACT making the most recent referral in their Animal Welfare Legislation Amendment Bill 2019 explanatory statement (https://www.legislation.act.gov.au/View/es/db_60140/20190926-72294/PDF/db_60140.PDF). We have looked for any UK literature or legislative documents stating that the AWA 2006 UK refers to the five domains and we have been unsuccessful. Would the reviewer kindly refer us to any papers or legislative documents for us to discuss and reference in this manuscript? We have however, included a reference to the five domains model on line 494.

l 610 I think the idea of sentencing being a deterrent has yet to be proved but this assertion needs references if any are available - I doubt anyone would think twice about being cruel to an animal because of the sentence unless it was a business decision eg illegal puppy farming, transport of animals were it may be a deterrent

You are correct, this was an error on our part. Thank you for picking that up we have changed it to “a symbolic gesture to show society the seriousness of the offence” (line 629), as this better reflects our intended meaning.

l 629 is there any data on recidivism in the different states relating to animal offence or the level of penalty that the State has eg how many times has the maximum sentence been applied in each State

Unfortunately, no, it is quite a big limitation in Australia as enforcement is done independently in each jurisdiction and as animal welfare cases are in the lowest courts (Magistrates Court) the information is not publicly available. Therefore, we do not have a system of national data collection to compare such data. We have briefly mention on line 760 that national data collection is required.

686 is there any evidence that either legislation (QLD or NSW) has had more impact on animal exhibits eg they tend to be based on one state as the rules are more law there or dont travel to a state due to the laws there

Again, there is no such evidence, however this is a very interesting point. It could also be the case that there are more animal exhibits in those jurisdictions and therefore the government has created specific legislation for that sole purpose, as QLD for example has many wildlife parks.

l 724 it would be useful to have data from each State on convictions per capita to show impact or otherwise of the different legislations on the animal cruelty offences eg is there less cruelty per capita in a State with a more modern approach to how the law is written or with mens rea included or not 

We completely agree – this would be great. However, the data does not exist unfortunately in Australia and is a huge limitation due to the independent nature of animal law enforcement by NGO and government agencies.

Reviewer 2 Report

The authors are to be commended on their research project and the execution. This study is novel, well-planned and the results will be of interest to academics and policy analysts alike. 

I agree that the decision not to include an analysis of codes and regulations is a significant limitation. That is especially so since many of those codes act as models for the Commonwealth (and thus could provide some degree of insight into the benefits of harmonisation). Nevertheless, I accept the authors' rationale for excluding them.

I did not see any reference to the proposed reforms in Victoria (https://engage.vic.gov.au/new-animal-welfare-act-victoria) and I wonder whether it might be worth mentioning these - hopefully this article will be published in time for it to be taken into account during that reform process.

Again, commendations to the authors for an innovative and useful study. A minor proof is all that is necessary from my perspective and I recommend it for publication.

Author Response

The authors are to be commended on their research project and the execution. This study is novel, well-planned and the results will be of interest to academics and policy analysts alike. 

Thank you very much.

I agree that the decision not to include an analysis of codes and regulations is a significant limitation. That is especially so since many of those codes act as models for the Commonwealth (and thus could provide some degree of insight into the benefits of harmonisation). Nevertheless, I accept the authors' rationale for excluding them.

Thank you. We do plan on doing a review on delegated legislation following similar methodologies to this manuscript. We are really looking forward to comparing those findings to the findings in this manuscript.

I did not see any reference to the proposed reforms in Victoria (https://engage.vic.gov.au/new-animal-welfare-act-victoria) and I wonder whether it might be worth mentioning these - hopefully this article will be published in time for it to be taken into account during that reform process.

Yes, we have now added references to these proposals specially discussing sentience on line 444-449, cruelty examples on lines 231-235, and miscellaneous Acts on lines 707-715

Again, commendations to the authors for an innovative and useful study. A minor proof is all that is necessary from my perspective and I recommend it for publication.

Thank you. We really appreciate the feedback.

Reviewer 3 Report

Overall

This is an excellent paper and certainly worthy of publication. More work in this area is needed, so the authors should be congratulated for their significant efforts. The method, results and discussions are well supported and explained.

However, the authors need to decide whether this manuscript is published in ANIMALS and revised to be more internationally relevant, or to publish it in an Australian focused law journal (i.e. ALJ). For the latter option, the manuscript is fine for publication.

Should the manuscript be considered for publication in ANIMALS which has a global readership, then minor/moderate revisions are suggested to ensure key international literature has been integrated and make it more relevant to a wider audience.

Inquiries and Court Rulings relating to Enforcement of Animal Welfare Law – needed

There also has been numerous reviews of animal protection law and frameworks across the Commonwealth, including the University of Otago Report https://ourarchive.otago.ac.nz/handle/10523/9276 and numerous parliamentary inquiries into animal welfare enforcement by charities (i.e. SPCA/RSPCA) including those in Western Australia, Victoria, United Kingdom and Ontario. The Ontario courts ruled that having the SPCA charity as an enforcement agency on behalf of the government was deemed unconstitutional, and as a result such functions were removed from this charity. There is a conflict with SPCA organisations providing enforcement but on the other hand working with meat producers to gain commercial benefit from accreditation schemes. Likewise primary industry departments are often conflicted with having to carry out livestock animal welfare, but also act under influence of trade, social, economic and political pressures. The recent landmark Judicial Review by the NZ High Court around Sow crates is a classic example why independent animal welfare enforcement is needed. http://nzala.org/w/wellington-high-court-finds-farrowing-crate-minimum-standards-unlawful/

The Otago Report also recommends (as other similar reports and political party manifestos do), that an independent authority be established to oversee animal welfare at the national level, namely an Independent Commissioner for Animal Welfare. This relates to 4.4.3 in the article. Following bushfires and floods in Australia, both Victoria and Queensland respectively have both created the office of the Inspector-General Emergency Management. This could be a model to contrast options for animal welfare at the federal level.

Some discussion around these developments and offering critical evaluation of the status quo would add value.

Five Domains – highly recommended

The paper refers frequently to animal welfare, but does not explore the core principals of animal welfare as laid out in the “five domains” (formerly five freedoms) model, of which many animal protection laws around the world are becoming based upon. Refer/add/discuss below article:
https://www.mdpi.com/2076-2615/10/10/1870

Climate Change & Disasters – highly recommended

According to Sawyer and Huertas (2018, p.2) 40 million animals are affected by disasters, and billions of animals are vulnerable to the risk of disaster impacts. The recent Australian bushfires has led to an estimated 500 million animals being killed. In 2017, the flooding of the township of Edgecumbe (NZ) in New Zealand (pop 1,500 approx.) resulted in over 1,000 animals left behind following evacuation that required to be rescued. In contrast for the entire year, the SPCA would have only taken 500 animals into possession (seized) – so disasters are starting to have more negative impacts, yet traditional animal welfare metrics have not often considered these significant impacts. With climate change likely to increase the frequency and consequence of weather related disasters (as well as urbanisation and population increase – both animal and human), the importance of animal protection (including livestock, wildlife, companion animals etc) and associated laws is going to become increasingly important.

A short commentary that no Australian state has specific animal disaster specific laws like those passed in the USA in the wake of Hurricane Katrina. The following articles are suggested for inclusion and discussion (including critique):

Glassey, S. Legal complexities of entry, rescue, seizure and disposal of disaster-affected companion animals in New Zealand. Animals 2020, 10, 1–12, doi:10.3390/ani10091583.

White, S. Companion Animals, Natural Disasters and the Law: An Australian Perspective. Animals 2012, 2, 380–394, doi:10.3390/ani2030380.

A major report into animal welfare emergency management presented to the NZ Parliament may also provide additional context for the paper. www.animalevac.nz/lawreport/

The recent live export disasters including the Queen Hind and Gulf Livestock 1, may also give discussion to international maritime law, as well as the legal obligations of masters of ships as persons being in charge or control.

International Conventions – highly recommended

With an international context, it would be beneficial to have the Australian laws discussed against international treaties and conventions (proposed):

https://www.animallaw.info/treaty/international-convention-protection-animals#:~:text=Summary%3A,and%20protection%20from%20cruel%20treatment.

https://www.animallaw.info/article/international-treaty-animal-welfare-0

https://digitalcommons.law.msu.edu/cgi/viewcontent.cgi?article=1466&context=facpubs

Inconsistency - suggested

The article frequently refers to inconsistencies and fragmentation, however it is unclear why this is an issue and if any occurrences of trans-border inconsistencies have caused negative animal welfare impacts (to play devils advocate).

Sentience - suggested

The role of sentience is another core philosophy of animal welfare. It has been recognised in other countries such as in the Animal Welfare Act 1999 (New Zealand), and currently in Victoria the government is reviewing POCTA and proposing to include sentience into the proposed replacement act https://engage.vic.gov.au/new-animal-welfare-act-victoria . A comment about the pending review would be timely.

Sociozoolgical Scale - suggested

Arluke and Sanders “sociozoological scale” concept is something that compliments both the philosophy around sentience and the five domains and worth integration to diversify the lens of the study away from strictly a legal perspective. The move to intensify animal production (i.e. factory farms) further increases the vulnerability of such animals when compared to traditional pasture based farming.

Finally

Again, this is a critically important paper that needs to be published. It is well constructed and presented. With some minor additions and discussion, it will become more useful for audiences outside of Australia. Well done! 

Author Response

Overall

This is an excellent paper and certainly worthy of publication. More work in this area is needed, so the authors should be congratulated for their significant efforts. The method, results and discussions are well supported and explained.

However, the authors need to decide whether this manuscript is published in ANIMALS and revised to be more internationally relevant, or to publish it in an Australian focused law journal (i.e. ALJ). For the latter option, the manuscript is fine for publication.

Should the manuscript be considered for publication in ANIMALS which has a global readership, then minor/moderate revisions are suggested to ensure key international literature has been integrated and make it more relevant to a wider audience.

Thank you for the praise and feedback. We have tried to include as much international literature as possible, as you have kindly referred us too. However, there are some limitations being that law is a has its own unique nuances based on jurisdiction. We also employed strict methodology with set inclusion and  exclusion criteria for the domestic law. If we did not apply the same strict standards to the international law we would be at risk of biasing our selection. The reasoning behind selecting an international journal is that many common law countries should relate to the principles discussed in this manuscript, and we hope that it can encourage other academics and policy makers to consider and critique the animal law framework in their home country, just as we have here. 

Inquiries and Court Rulings relating to Enforcement of Animal Welfare Law – needed

There also has been numerous reviews of animal protection law and frameworks across the Commonwealth, including the University of Otago Report https://ourarchive.otago.ac.nz/handle/10523/9276 and numerous parliamentary inquiries into animal welfare enforcement by charities (i.e. SPCA/RSPCA) including those in Western Australia, Victoria, United Kingdom and Ontario. The Ontario courts ruled that having the SPCA charity as an enforcement agency on behalf of the government was deemed unconstitutional, and as a result such functions were removed from this charity. There is a conflict with SPCA organisations providing enforcement but on the other hand working with meat producers to gain commercial benefit from accreditation schemes. Likewise primary industry departments are often conflicted with having to carry out livestock animal welfare, but also act under influence of trade, social, economic and political pressures. The recent landmark Judicial Review by the NZ High Court around Sow crates is a classic example why independent animal welfare enforcement is needed. http://nzala.org/w/wellington-high-court-finds-farrowing-crate-minimum-standards-unlawful/

The Otago Report also recommends (as other similar reports and political party manifestos do), that an independent authority be established to oversee animal welfare at the national level, namely an Independent Commissioner for Animal Welfare. This relates to 4.4.3 in the article. Following bushfires and floods in Australia, both Victoria and Queensland respectively have both created the office of the Inspector-General Emergency Management. This could be a model to contrast options for animal welfare at the federal level.

Some discussion around these developments and offering critical evaluation of the status quo would add value.

We chose to review the statutes independently from their subsequent enforcement models for two reasons: (1) to keep the length of the manuscript at a readable level, if we were to also critique enforcement models and the inquiries it would increase the page numbers substantially to an already substantially long manuscript, and (2) earlier this year we actually published a paper and discussed these inquires and enforcement models in great detail (https://www.mdpi.com/2076-2615/10/3/482). In section 4.4.3 (lines 750-759) we have included references to these inquires and reports now as per your recommendation, to inform the reader of their existence and made further reference to our previous paper where we discussed them in greater detail. Thank you for this comment.

Five Domains – highly recommended

The paper refers frequently to animal welfare, but does not explore the core principals of animal welfare as laid out in the “five domains” (formerly five freedoms) model, of which many animal protection laws around the world are becoming based upon. Refer/add/discuss below article:
https://www.mdpi.com/2076-2615/10/10/1870

Australian animal welfare legislation is yet to recognise Mellor’s Five Domains model, as the five freedoms are still regularly discussed in parliamentary debate documents, with ACT making the most recent referral in their Animal Welfare Legislation Amendment Bill 2019 explanatory statement (https://www.legislation.act.gov.au/View/es/db_60140/20190926-72294/PDF/db_60140.PDF). We have looked for any international literature or legislative documents stating that animal protection statutes refer to the five domains and we have been unsuccessful. We have however included a reference to the provided article on line 494, stating that the developing domains model is showing the progression animal welfare science is making.

Climate Change & Disasters – highly recommended

According to Sawyer and Huertas (2018, p.2) 40 million animals are affected by disasters, and billions of animals are vulnerable to the risk of disaster impacts. The recent Australian bushfires has led to an estimated 500 million animals being killed. In 2017, the flooding of the township of Edgecumbe (NZ) in New Zealand (pop 1,500 approx.) resulted in over 1,000 animals left behind following evacuation that required to be rescued. In contrast for the entire year, the SPCA would have only taken 500 animals into possession (seized) – so disasters are starting to have more negative impacts, yet traditional animal welfare metrics have not often considered these significant impacts. With climate change likely to increase the frequency and consequence of weather related disasters (as well as urbanisation and population increase – both animal and human), the importance of animal protection (including livestock, wildlife, companion animals etc) and associated laws is going to become increasingly important.

A short commentary that no Australian state has specific animal disaster specific laws like those passed in the USA in the wake of Hurricane Katrina. The following articles are suggested for inclusion and discussion (including critique):

Glassey, S. Legal complexities of entry, rescue, seizure and disposal of disaster-affected companion animals in New Zealand. Animals 2020, 10, 1–12, doi:10.3390/ani10091583.

White, S. Companion Animals, Natural Disasters and the Law: An Australian Perspective. Animals 2012, 2, 380–394, doi:10.3390/ani2030380.

A major report into animal welfare emergency management presented to the NZ Parliament may also provide additional context for the paper. www.animalevac.nz/lawreport/

The recent live export disasters including the Queen Hind and Gulf Livestock 1, may also give discussion to international maritime law, as well as the legal obligations of masters of ships as persons being in charge or control.

This is a very interesting and critically important topic. However, we are unsure it fits the scope of this manuscript as we followed strict inclusion criteria  (refer to line 82), and any such acts would thus have been excluded based on our methods. We have included a reference to emergency management statutes as well as the most recent publication provided on line 100 to inform the reader why we excluded these types of laws as well as referring them on to the publication for further commentary, as we do acknowledge that this is an important topic.

International Conventions – highly recommended

With an international context, it would be beneficial to have the Australian laws discussed against international treaties and conventions (proposed):

https://www.animallaw.info/treaty/international-convention-protection-animals#:~:text=Summary%3A,and%20protection%20from%20cruel%20treatment.

https://www.animallaw.info/article/international-treaty-animal-welfare-0

https://digitalcommons.law.msu.edu/cgi/viewcontent.cgi?article=1466&context=facpubs

Animal welfare law in Australia is domestic (state based) due to Constitutional restrictions, thus any treaties or conventions would only be included in federal law and have minimal impact on the animal welfare statutes which are at state and territory level, given that enforcement is also carried out independently in each jurisdiction. We have however, included a brief discussion on OIE and the fact that Australia along with many other countries is a member. Refer to lines 485-489.

Inconsistency - suggested

The article frequently refers to inconsistencies and fragmentation, however it is unclear why this is an issue and if any occurrences of trans-border inconsistencies have caused negative animal welfare impacts (to play devils advocate).

We discussed the issue of no interstate recognition of prohibition orders from an inconsistency perspective and included a specific example of this problem on lines 645-657 (reference 343). However, our overall conclusion on line 768 was that these laws were broadly consistent across each jurisdiction and only had minor inconsistencies/shortcomings (eg unclear animal definitions, reliance on punishment methods to promote a positive concept, alternative penalties, interstate recognition of court orders and recognition of animal sentience). We concluded that due to Constitutional restrictions a uniform approach would not be feasible in Australia, especially without the introduction of a unified enforcement model. We hope this clears this comment up.

Sentience - suggested

The role of sentience is another core philosophy of animal welfare. It has been recognised in other countries such as in the Animal Welfare Act 1999 (New Zealand), and currently in Victoria the government is reviewing POCTA and proposing to include sentience into the proposed replacement act https://engage.vic.gov.au/new-animal-welfare-act-victoria . A comment about the pending review would be timely.

We have made reference to the proposals for the Vic POCTA Act to the existing sentience discussion from lines 444-449.

Sociozoolgical Scale - suggested

Arluke and Sanders “sociozoological scale” concept is something that compliments both the philosophy around sentience and the five domains and worth integration to diversify the lens of the study away from strictly a legal perspective. The move to intensify animal production (i.e. factory farms) further increases the vulnerability of such animals when compared to traditional pasture based farming.

This is a very interesting concept, however as this study followed a methodological approach to review statutes with a strict legal focus. We feel that this would be straying too far from the research question of the review.

Finally

Again, this is a critically important paper that needs to be published. It is well constructed and presented. With some minor additions and discussion, it will become more useful for audiences outside of Australia. Well done!

Thank you very much for taking the time to read and review the paper. Your comments have been very valuable.

This manuscript is a resubmission of an earlier submission. The following is a list of the peer review reports and author responses from that submission.

Round 1

Reviewer 1 Report

This is an excellent overview of the problems of inconsistency in animal welfare laws across jurisdictions in the same country. It should be of interest to readers in many countries. The problems are particularly true in the U.S. where we have 50 definitions of “animal” and “cruelty” (not counting D.C., Guam, US Virgin Islands  and Puerto Rico). The authors might gain some additional insights to the issues they raise by reviewing an analysis of that problem in:

Arkow, P. and R. Lockwood.  2016. Defining animal cruelty, in. C.L. Reyes and M. Brewster (Editors) Animal Cruelty: A Multidisciplinary Approach to Understanding. 2nd Edition. Durham, NC: Carolina Academic Press, 3-23.

Although the authors did not include animal fighting in their analysis these ”focus on the actions of the person rather than implications to the animal”, I would strongly suggest that these laws be included. Dogfighting in particular is a major welfare concern since the crime invariably involves animal cruelty. However, since the penalties for animal fighting are usually more severe than those for animal cruelty, many - if not most- prosecutions for animal fighting do not include animal cruelty charges since these may require more effort to prove (e.g. needing veterinary forensic reports for each animal).

I would like to see some discussion of any exemptions in these laws - beyond the usual exclusions of common agricultural or veterinary practices. Several US state laws have special exemptions for rodeos, animal in parades (Louisiana) etc. I would be interested to see if Australia has any such provisions.

Brief mention is made of alternative sentencing including psychological assessment and treatment. Although these are not direct animal welfare provisions, they play an important potential role in the prevention of cruelty. US states vary widely in the provisions for such assessment and treatment and, as the authors note, legislators are just beginning to recognize this as an important component of responding to animal hoarding.

The authors make reference to appreciation of the connections between animal cruelty and interpersonal violence in shaping laws. I would like to see some commentary on whether any of the Australian cruelty laws recognize this - e.g. in the form of elevated charges if the act is committed in the presence of a minor (a common provision in US state laws) or acts done to coerce or intimidate a person, e.g. in interpersonal violence. While not directly related to animal welfare, such provisions are a good indicator of the system’s recognition of the added impact of cruelty on those who are attached to the animal victim and often involve significant elevation of penalties.

Author Response

This is an excellent overview of the problems of inconsistency in animal welfare laws across jurisdictions in the same country. It should be of interest to readers in many countries. The problems are particularly true in the U.S. where we have 50 definitions of “animal” and “cruelty” (not counting D.C., Guam, US Virgin Islands  and Puerto Rico). The authors might gain some additional insights to the issues they raise by reviewing an analysis of that problem in:

Arkow, P. and R. Lockwood.  2016. Defining animal cruelty, in. C.L. Reyes and M. Brewster (Editors) Animal Cruelty: A Multidisciplinary Approach to Understanding2nd Edition. Durham, NC: Carolina Academic Press, 3-23.

Thank you for providing this reference, it has provided us with some insights into U.S. animal protection law. We have included this work on lines 577-584 of the manuscript.

Although the authors did not include animal fighting in their analysis these ”focus on the actions of the person rather than implications to the animal”, I would strongly suggest that these laws be included. Dogfighting in particular is a major welfare concern since the crime invariably involves animal cruelty. However, since the penalties for animal fighting are usually more severe than those for animal cruelty, many – if not most- prosecutions for animal fighting do not include animal cruelty charges since these may require more effort to prove (e.g. needing veterinary forensic reports for each animal).

We do agree with this comment, therefore we have included a table of offences pertaining to prohibited activities and items on lines 239-251.

I would like to see some discussion of any exemptions in these laws – beyond the usual exclusions of common agricultural or veterinary practices. Several US state laws have special exemptions for rodeos, animal in parades (Louisiana) etc. I would be interested to see if Australia has any such provisions.

In Australia most of the exemptions to offences occur in the delegated legislation of Regulations and Codes of Practice. We did not include these in this paper as we think a review of this legislature requires its own separate manuscript. We have included a discussion on this delegated legislation on lines 536-548 and alluded to the need for further discussions on this issue. As in the US, this is also an important issue in Australian animal protection legislation.

Brief mention is made of alternative sentencing including psychological assessment and treatment. Although these are not direct animal welfare provisions, they play an important potential role in the prevention of cruelty. US states vary widely in the provisions for such assessment and treatment and, as the authors note, legislators are just beginning to recognize this as an important component of responding to animal hoarding.

Thank you for this comment, we agree whole-heartedly.

The authors make reference to appreciation of the connections between animal cruelty and interpersonal violence in shaping laws. I would like to see some commentary on whether any of the Australian cruelty laws recognize this - e.g. in the form of elevated charges if the act is committed in the presence of a minor (a common provision in US state laws) or acts done to coerce or intimidate a person, e.g. in interpersonal violence. While not directly related to animal welfare, such provisions are a good indicator of the system’s recognition of the added impact of cruelty on those who are attached to the animal victim and often involve significant elevation of penalties.

In Australia, such provisions are commonly included in the state-based Domestic Violence Acts, where the impacts of animal cruelty are recognised as a form of domestic abuse. We mentioned this briefly on line 119, as a part of our exclusion criteria. However, we also included a discussion on aggravating factors on lines 663, where factors that contribute to culpability (such as acting in groups or lack of remorse in Australia) often receive a penal elevation.

Reviewer 2 Report

It is a well done paper.  One might quarrel with the force of the conclusions drawn about the lack of any need for a general federal law.  As the authors note, enforcement patterns and jurisprudential variation across states may lead to considerably less uniformity than a cold review statutory text suggests.  It is also not clear why uniformity at the state level, if true, suggests that a federal law that might promote even greater uniformity is not still a good idea.  Surely a single federal standard would promote greater uniformity than relative uniformity that exists across jurisdictions and could change at any moment during any legislative session, no?  

Author Response

It is a well done paper.  One might quarrel with the force of the conclusions drawn about the lack of any need for a general federal law.  As the authors note, enforcement patterns and jurisprudential variation across states may lead to considerably less uniformity than a cold review statutory text suggests.  It is also not clear why uniformity at the state level, if true, suggests that a federal law that might promote even greater uniformity is not still a good idea.  Surely a single federal standard would promote greater uniformity than relative uniformity that exists across jurisdictions and could change at any moment during any legislative session, no? 

It is true that a federal standard would be the absolute best-case scenario for animal protection legislation in Australia, especially given the movement of animals across jurisdictions and the benefits of cross-jurisdictional data collections. However, we believe that in Australia at least, a federal uniform approach would not be feasible. We do not think that the level of inconsistencies we found from this review are great enough to warrant a uniform approach, especially when enforcement will still be carried out by state-based authorities. We have tried to take a pragmatic approach, and considering that crimes statutes and domestic violence statutes are state/territory-based in Australia, we don’t believe that animal welfare is enough of a social issue that a major reform such as federal uniformity (given our Constitutional restriction in Australia), will be achievable. Theoretically it would be great, but realistically, it would not be feasible in the foreseeable future. We have tried to make this clearer in the conclusion on line 848.

Reviewer 3 Report

Please see detailed referee report attached. 

Author Response

This is a well-presented and well-written paper, which reads well. The premise of the article, namely to provide a ‘comprehensive legislative review relating to governance of animal welfare that can serve as a basis for guiding future discussions around need for a uniform approach to animal welfare law’ [Lines 60-61] is a laudable one. The authors are correct in arguing that a lack of Commonwealth leadership and harmonisation across state and territory approaches to animal welfare has often been described as a challenge to the achievement of good animal welfare outcomes. However, I do not think that this study provides the claimed comprehensive review of relevant applicable law, for the reasons I set out below. Nonetheless, I do think the paper presents a useful comparison of key aspects of animal welfare laws across Australian states and territories at what might be described as a higher order or overview level. This is valuable in itself, provided the authors are more explicit as to the limitations of what they review and the implications of those limitations upon conclusions that can be drawn. If they do so, I believe the paper is a welcome addition to the literature in this field and congratulate the authors on their important work.

Thank you for this comment. We have removed the wording of “comprehensive review” to just “review” throughout the manuscript, as we understand focusing singularly on statutes creates limitations and it was not our intention to present that these are the only problems within animal protection legislation.

There are three main areas where I believe the authors need to strengthen the paper: 1) This is not a comprehensive review – the paper must better acknowledge the limitations of the study due to the sources of law and principles left out and the implications of doing so for the conclusions that can be drawn I am not convinced around the justifications for the omission of certain legislative instruments and secondary legal sources from the study. For example, the omission of regulations (many of which contain substantive cruelty offences that are accompanied by clear prescribed penalties) and of Codes of Practice and Standards (the latter of which are enforceable in some jurisdictions through a complex relationship with primary legislation) is significant. While their legal form may be different to primary acts of Parliament, it is not clear how in substance this difference is significant. While the authors are correct that codes of practice are not directly enforceable (unlike offences contained in secondary legislation and the legal status of Standards in some states), they do constitute part of the content of what is or is not a punishable act of cruelty under state and territory animal welfare laws. Legally this either operates in terms of their providing a defence to cruelty offences or by precluding the application of cruelty provisions entirely. The latter legislative approach in particular tends to mirror the effect of a more narrow or wide definition of ‘animal’, which is a subject given a treatment in this study as relevant and important. Indeed, similarities or differences across primary legislation in each state and territory in terms of how codes/standards are related to/integrated into animal welfare/anti-cruelty laws seems an integral feature of the legislative picture and should be subject to its own table of comparison.

We agree with this comment and understand that excluding delegated legislation is a substantial limitation to this paper. However, as we have investigated animal protection statutes within multiple subject domains (animal welfare, wildlife, crimes, fish, etc), if we were to investigate the regulations and codes as well, we would need to include them for all 42 statutes included. As you have acknowledged, this would be too large of a task for a journal article, however, we have made revisions to lines 74, 418-424, 812-818 and 856 to better acknowledge this limitation, and allude to the need for further research within this domain.

It also seems noteworthy that codes and standards have been the subject of precisely the kind of significant national harmonisation efforts the authors refer to as the baseline of their study, for the very reason that they are particularly impactful upon the real world welfare conditions of animals and how those are regulated under and through Australian welfare law. I appreciate it would be a major revision to include these sources (regulations, codes, and standards) in this study and potentially far too big a task for a journal article. However, I think the paper requires a dedicated discussion of how regulations, codes and standards do have a close link to the enforceable animal welfare legal system and to acknowledge what is left out more clearly. This is crucial so that a non-specialist reader can understand the limitations of the method more fully.

We have included some discussion on the relationship between regulations and codes on lines 418-424 and 812-818.

In my view, mention of the legal status of Codes (which first occurs at Line 417 in the context of a discussion of ‘legally recognised sentience’) comes too late. Another location where the implications of the decision to omit regulations, codes and standards from the study invites further elaboration is Part 3.6 Miscellaneous Statutes. This is because the reason that some states or territories do not have specific Acts on certain issues (exhibited animals; racing etc) is precisely because they are contained in other legal sources. The authors may indeed wish to reflect on how the difference in approach has implications for animal welfare in those industries across jurisdictions.

As we have tried to adhere to scientific writing principles, where the results describe the tables and the explanation occurs in the discussion, we included the relationship between the miscellaneous statutes and use of delegated legislation on lines 792-810.

I am similarly not convinced about the justification for excluding prohibited procedures and prohibited tools. This is explained (at Lines 206-207) on the basis that these offences tend to focus upon the action of the offender rather than the nature of the harm to the animal. However, this is not all that compelling because a) offences described in this way are almost always based upon the impact of the relevant activity on the animal – so the distinction again seems to be one of form over substance and b) some basic offences are likewise described in this way (e.g. abandonment of animals; failure to provide treatment; overworking; bestiality offences) and would therefore, on this logic, also be excluded. Moreover, some prohibited procedure offences do refer to the impact upon the animal as part of the offence – see for example the Victorian Prevention of Cruelty to Animals Regulations – Regulation 34 (on prohibited confinement traps) and Regulation 83 (on poking/harming animals at a rodeo).

Thank you for this comment. We have now included a table in the results section comparing prohibited activities and items cross-jurisdictionally on lines 239-251.

Again, it may be possible for the authors to simply justify their project as a limited (not comprehensive) comparison of Australian animal welfare legislation (limited to Acts of Parliament) and that the project does not cover the other regulatory instruments and miscellaneous prohibited procedures etc – and that these are perhaps matters for further study – unless they can find a better way to justify their exclusion. Again, I think it is still useful to provide the study as is, provided the reader is alerted to what is missing and how this plays a role in any assessment of the adequacy or otherwise of a state-centric regulatory response to welfare.

We have removed the wording of “comprehensive” from this review and made revisions on lines 418-424, 812-818 and 856 acknowledging the limitations of this study and stating the need for future research. We welcome this critique and hope these edits suffice.

2) The role of the common law in the animal welfare legal environment needs elaboration I think it is necessary for the authors to better address the relationship between common law and legislation in the Australian animal welfare legal framework. A number of key concepts discussed in this paper either originate from the common law, or are subject to substantial elaboration in the common law (e.g. person in charge; unreasonable suffering). In a couple of places the authors mention the role of the common law and of precedent (see for example Lines 224-6 and 538- 546]. However, I do not think the analysis provided is adequate. The role of the common law is a key feature of the Australian legal system that serves to provide a common source in the interpretation and application of distinct state and territory welfare acts. Given this is crucial to the purported focus of the paper – whether and to what degree national uniformity driven by reform at the Commonwealth level is needed – I think more depth of discussion is warranted than simply referring to the doctrine of precedent in general terms. Just a bit more on how and when common law concepts are authoritative/ influential in the interpretation of distinctive state and territory legislation where similar language is employed would be worthwhile (see the difference, for example, between the role of common law when a court is interpreting uniform legislation versus other types of state/territory legislation).

We do agree with this comment. We have included on lines 418-424 and 812-818 that common law was excluded from this paper and introduced the principles to the reader, acknowledging that legislation and common law work symbiotically in Australia. We have reiterated that this paper only focused on the black letter of statutes and that common law will play an important role in applying the principles discussed in this paper.

3) The discussion (Part 4) needs to better align with the purpose of the study (Parts 1 and 2) and findings (Part 3) While interesting, the discussion section of the paper often does not link to the purported focus of the study or to the method adopted, but considers distinct normative debates (e.g. the deterrent function of law; the value of a legal statement of sentience; issues of enforcement). Ideally, this part of the paper should connect back to what 3 the authors have stated is the primary function of the study – to determine whether the demand for national uniformity or national legislative reform is warranted – and the substantive offence provisions reviewed and compared. For example, the discussion of differences in definitions of animals across states and territories (4.1.1) could be better linked to the study undertaken. While there may be reasons why legislation is informed not only by science but also by the purposes of a given law, do the authors think this explains and justifies the differences that exist across states and territories? Alternatively, do the differences remain problematic and why or why not? Likewise, the discussion of differences in wildlife statutes is interesting but it is not quite clear what the reader should take away on the core issue of the call for harmonisation across states and territories.

The discussion was designed to focus on specific aspects that we found from our analysis, where we found that the majority of the shortcomings in animal protection legislation was actually consistent throughout the statutes in each subject domain. In regard to the animal definition within animal welfare statutes, we explained why there may be differences within the definition and concluded that it requires revision to better reflect animal sentience. In regard to the wildlife definition discussion, this was to show the reader that definitions are dependant on the objective of the statute, and that there are differences between different subject domains. We made revisions to the introduction on lines 76-78 to inform the reader on the issue we are discussing in the introduction.

The discussion on legal recognition of animal sentience (4.1.2) seems out of place and is, in fact, principally addressing a different issue (the differential treatment of animals depending upon social context). I think the discussion on how codes of conduct result in animals being treated in ‘anthropocentric’ ways, needs to come earlier and be addressed to the limitations of this study (as outlined above) and perhaps linked to the value of a companion or further study that focuses explicitly on omitted sources. I would suggest the authors otherwise omit or reduce the section on ‘legal recognition of sentience’. If kept, the authors should be less ambivalent as to whether the inclusion of such recognition – such as in the purposes section of the Australian Capital Territory law- will have more than symbolic implications. This could be informed by reference to relevant Australian statutory interpretation principles rather than via a comparison to the quite different European legal context.

This discussion was included as an inconsistency was found with ACT being the only state/territory to include sentience in their legislation. This section was to follow on from the conclusion that the animal definitions should be reformed to coincide with animal sentience and that ACT (with the most inclusive animal definition) actually went a step further to acknowledge sentience. We have made revisions to this section to better articulate this concept and acknowledge the interpretation legislation in Australia, which may guide this sentience recognition further (lines 455-).

The point regarding a lack of need for aggravated offences given discretion in sentencing seems rather remarkable [Lines 560-564]. It fails to account for how sentencing discretion is framed by the relevant offence and penalty attached – so I am not sure I would agree that the inclusion of aggravated offences in primary legislation is simply about messaging. More explanation is needed to justify the position taken by the authors or the claim should be more modest – that the lack of aggravated offences in some states and territories is somewhat mitigated by the discretion in sentencing afforded to judges. To say more than that in my view requires some empirical review of actual sentencing practice comparing those states with aggravated offences and those without.

We have removed this statement as we understand that our meaning may be misinterpreted.

Finally, it is not clear why the authors chose to raise the issue of distinct enforcement agencies in the context of their discussion of bestiality offences rather than elsewhere. This issues seems a broader one and (as the authors state) largely beyond the method and purpose of this study. If the authors wish to include it, it seems to be a standalone issue that should be informed by a comparison of the relevant enforcement provisions across the states and territories in the legislative instruments surveyed.

We agree with this comment and have removed this section.

Other specific suggestions: Line 40 – I do not think it is accurate to describe the Australian Constitution as ‘providing’ residual powers to the states – rather it operates such that there are areas of residual power exercised by states. Likewise, I do not agree that the issue of lack of Commonwealth power to deal with animal welfare can be explained by the property status of animals – indeed the Commonwealth does exercise power over property where, for example, it comes within an existing head of power such as the trade and commerce power. I therefore think those two issues (the legal status of animals and the Constitutional powers of states/territories versus the Commonwealth) should be decoupled.

We agree that this was poor choice of wording. We have revised this sentence to acknowledge that the lack of inclusion of animal welfare in the Constitution renders animal welfare as a residual power in the domains of the states/territories and removed the statement of the property status (lines 42-44).

Table 5 – it could be clearer whether the penalties listed reflect those applicable to a natural person or a body corporate. Indeed, the fact and how this distinction affects penalties across jurisdictions could form part of the comparison.

All penalties included in this review were applicable to a natural person. We have included this information in the table legends and stated that different penalties are applicable to body corporates.

Table 6 – I find it curious that the Victorian provisions where the offence includes a comparable notion of unreasonable pain and suffering are not included in the table (for example, Art 6(1)(b), 6(1)(c), 6(1)(d), 6(1)(e), 36). Again, the method of comparison seems to focus on matters of form rather than substance.

The Victorian Sentencing Council has stated that there is no cruelty definition included in the Prevention of Cruelty to Animals Act 1986, in that the Act only includes examples of what cruelty constitutes. We have reviewed the table to acknowledge that Victoria only included example of cruelty (which does include unreasonable suffering), but no definition. We hope this suffices.

Table 7 – is not that informative, given the need for the reader to cross-reference to the endnotes to find the relevant information. The authors may wish to consider adding the relevant provision within the table itself, as they do in Table 5.

We have included the provisions in Table 7.

Line 260 – grammatical error where it states ‘are the similar as those’

Thank you, this have been rectified.

Line 390 – grammatical error ‘where deceased are’

Again, this has been rectified.